# Conservative classifiers do consistently well with improving agents: characterizing statistical and online learning

**Dravyansh Sharma**
Northwestern University
Toyota Technological Institute at Chicago

**Alec Sun**
University of Chicago

## Abstract

Machine learning is now ubiquitous in societal decision-making, for example in evaluating job candidates or loan applications, and it is increasingly important to take into account how classified agents will react to the learning algorithms. The majority of recent literature on *strategic classification* has focused on reducing and countering deceptive behaviors by the classified agents, but recent work of Attias et al. [5] identifies surprising properties of learnability when the agents genuinely improve in order to attain the desirable classification, such as smaller generalization error than standard PAC-learning. In this paper we characterize so-called *learnability with improvements* across multiple new axes. We introduce an asymmetric variant of *minimally consistent* concept classes and use it to provide an exact characterization of proper learning with improvements in the realizable setting. While prior work studies learnability only under general, arbitrary agent improvement regions, we give positive results for more natural Euclidean ball improvement sets. In particular, we characterize improper learning under a generative assumption on the data distribution. We further show how to learn in more challenging settings, achieving lower generalization error under well-studied bounded *noise* models and obtaining mistake bounds in realizable and agnostic *online* learning. We resolve open questions posed by Attias et al. [5] for both proper and improper learning.

## 1 Introduction

Suppose that a school is trying to create a machine learning classifier to admit students based on their test score. The school uses a cutoff $\hat{\theta}$ to determine whether a student should be admitted. If the school publishes this cutoff, then students immediately below the cutoff will want to boost their test scores in order to be admitted, for example by studying harder or by registering for booster courses. This is an example of where deploying a classifier can influence the behavior of the agents it is aiming to classify. We assume binary classification and that agents want to be positively classified.

There are two views one can take on this phenomenon. The first is that the actions that agents take in response to the deployed classifier do not truly improve the agent's quality. This setting is known as strategic classification [36] or measure management [16]. A second view, however, which is the focus of this work, is that agents' actions lead to real improvements. For example, studying or taking classes in order to get a higher exam score can genuinely make a student more qualified, so the school would want to admit any student that achieves their desired cutoff $\hat{\theta}$. This setting, when agents respond to the classifier to potentially improve their classification while changing their true features in the process, is known as *strategic improvements* [40, 45].

39th Conference on Neural Information Processing Systems (NeurIPS 2025).

Previous works seek to efficiently incentivize and maximize agent improvement [45, 40, 53]. In a recent paper, Attias et al. [5] instead take the agent improvement function as given and study *statistical learnability* with improvements. Attias et al. [5] show fundamental differences in the learnability of concept classes compared to both standard PAC-learning where the agents cannot respond to the classifier, as well as the strategic setting where the agent tries to deceive the classifier to obtain a more favorable classification. Surprisingly, learning with improvements can sometimes be easier than the standard PAC setting, and it can sometimes be harder than strategic classification. Attias et al. [5] provide concrete examples showing the separation between standard PAC-learning, strategic PAC-learning, and PAC-learning with improvements. Attias et al. [5]'s results focus on realizable and fully offline learning. They give necessary and sufficient conditions for proper learning in terms of the intersection-closed property of concept classes.

In this paper, we give an exact characterization of which concepts classes are PAC learnable in the proper and realizable setting using a new notion which we call *nearly minimally consistent* concept classes. This improves upon prior work by closing the gap between the necessary and sufficient conditions for proper, realizable PAC learning. We also design classifiers to maximize accuracy for strategic improvements in more challenging settings as well as under more natural assumptions. For improper learning, we show a natural sufficient condition on the data distribution for learning when the agents can improve within Euclidean balls. Under this condition, we obtain tight sample complexity bounds, up to logarithmic factors.

Next, we turn our attention to learning with improvements in more challenging settings. We study learning with bounded label noise (including random classification noise and Massart noise), where we can no longer rely on positive labels in our training set as being perfectly safe. For example, a weak student who studied a small random subset of the syllabus might get lucky with some small probability if the test questions happen to be from the part the student reviewed. We show how to design Bayes optimal classifiers, that is achieve $\mathsf{OPT}$ expected error in the learning with improvements setting. In contrast, one can only hope to achieve $\mathsf{OPT} + \varepsilon$ in standard PAC learning.

Finally, we initiate the study of online learning with improvements, which may be more realistic than assuming that the agents come from a fixed distribution. This is particularly important to study when agents can move in response to our classification over time. For example, a false positive in our published classifier may be exploited by an increasing number of agents. Further, we handle even the agnostic learning in the online setting where no classifier available to the learner may be perfect. We design "conservative" versions of the majority vote classifier which achieve near-optimal mistake bounds for both realizable and agnostic online learning with improvements.

**Contributions.** Our paper makes the following contributions on learning with improvements in challenging and natural settings:

- **Proper learning for any improvement function.** In Section 3, we introduce a new property of concept classes called *nearly minimally consistent* and show that this property fully characterizes which concept classes can be learned with improvements for any improvement function in the proper, realizable setting. This resolves an open question of Attias et al. [5].

- **Improper learning for Euclidean ball improvement sets.** In Section 4, we move beyond proper learning. We show that to get positive results in the improper setting we need to make assumptions about both the agent improvement set and the data distribution. We prove that the simple memorization learning rule can learn *any* concept class improperly, even those with infinite VC-dimension, assuming that the improvement set is or contains a Euclidean ball and that the data distribution satisfies a *coverability* condition. Both of these assumptions are natural and appear in prior literature. This addresses another question raised by Attias et al. [5].

- **Learning with noise.** In Section 5, we construct optimal algorithms for learning linear separators in the improvements setting with bounded label noise under isotropic logconcave distributions and instantiate our algorithms for many well-studied noise models. To the best of our knowledge, we are the first to consider learning with strategic agents under label noise.

- **Online learning on a graph.** In Section 6, we study mistake bounds for online learning with improvements for the discrete graph model of Attias et al. [5] in the more challenging online setting. In both realizable and agnostic settings, we prove that risk-averse modifications of the weighted majority vote algorithm enjoy near-optimal mistake bounds.

**Related work.** In each of the settings we study above, we discuss the conceptual differences between learning with improvements and learning under strategic classification. More generally, our paper is related to several works studying learning in strategic and adversarial environments, such as strategic classification [37, 46, 19, 1, 35], learning with improvements [40, 35, 5], reliable learning [49, 33], and learning with noise [10]. For a detailed discussion of related work, see Appendix A.

## 2  Model

Attias et al. [5] propose the following formal model for learning with improvements. Let $\mathcal{X}$ denote the instance space consisting of points or agents, and the label space is binary $\{0, 1\}$. Here the label $0$ is called the *negative* class and label $1$ is called the *positive* class, and all the agents would prefer to be classified positive. Let $\Delta : \mathcal{X} \to 2^{\mathcal{X}}$ denote an *improvement function* that maps each point $x \in \mathcal{X}$ to its *improvement set* $\Delta(x)$, to which $x$ can potentially move after the learner has published the classifier in order to be classified positively. For example, if $\mathcal{X} \subseteq \mathbb{R}^d$, a natural choice for $\Delta(x)$ could be an $\ell_2$-ball centered at $x$. Let $\mathcal{H} \subseteq \{0, 1\}^{\mathcal{X}}$ denote the concept space, that is, the set of candidate classifiers. We assume the existence of a ground truth function $f^* : \mathcal{X} \to \{0, 1\}$, which represents the true label of every point.

The goal is to learn the ground truth by sampling labeled instances from $\mathcal{X}$. After seeing several samples from $\mathcal{X}$, the learner will publish a classifier $h : \mathcal{X} \to \{0, 1\}$. Each point now reacts to $h$ in the following way: if the agent was classified negative by $h$, it tries to find a point in its improvement set $\Delta(x)$ that is classified positive by $h$ and moves to it. The agent will only move if such a positive point exists. We formalize this as the *reaction set* with respect to $h$,

$$\Delta_h(x) = \begin{cases} \{x\} & \text{if } h(x) = 1 \text{ or if } \{x' \in \Delta(x) \mid h(x') = 1\} = \emptyset \\ \{x' \in \Delta(x) : h(x') = 1\} & \text{otherwise.} \end{cases} \tag{2.1}$$

Note that if $h$ classifies $x$ as positive, $x$ stays in place. If $h$ classifies $x$ as negative, there are two cases. Either, there is no point in its improvement set where the agent is classified positive by $h$ and the agent does not move. Else, the agent moves to some point $x'$ to be predicted positive by $h$. Our improvement set model is equivalent to the behavior of utility-maximizing agents that have a utility equal to $h(x) - \text{cost}(x)$, and $\Delta(x) = \{x' \in \mathcal{X} \mid \text{cost}(x) < 1\}$. These agents have a utility of $1$ for being classified as positive, a utility of $0$ for being classified as negative, and incur a cost for moving, where $\Delta(x)$ corresponds to the points that $x$ can move to at a cost less than $1$.

A *classification error* in the improvements setting is said to occur if there exists a point in the reaction set of $x$ where $h$ disagrees with $f^*$, namely

$$\text{Loss}(x; h, f^*) = \max_{x' \in \Delta_h(x)} \mathbf{1} \left[ h(x') \neq f^*(x') \right]. \tag{2.2}$$

According to Eq. (2.2), agents $x$ with $h(x) = 0$ will improve to a point $x'$ in their reaction set with $h(x') = 1$ if possible, breaking ties in the *worst case* in favor of points $x'$ for which $f^*(x') = 0$. This definition is natural if we want our classifiers to be robust to unknown tiebreaking mechanisms, and would also make sense if improving to points whose true label according to $f^*$ is negative is less costly than improving to points whose true label is positive. Hence Eq. (2.2) implies that a classification error occurs for $x$ if one of the following is true: $x$ itself is a false positive, $x$ is a false negative and there is no positive $x' \in \Delta_h(x)$ that $x$ can improve to, or $x$ is negative and there exists a false positive $x' \in \Delta_h(x)$ that $x$ can move to. This loss function favors *conservative* classifiers that label uncertain points as negative rather than positive. For example, consider a scenario in which there are no false positives, equivalently $\{x \mid h(x) = 1\} \subset \{x \mid f^*(x) = 1\}$. Note that $h$ can have zero loss so long as all points $x$ for which $h(x) = 0$ and $f^*(x) = 1$ can improve to some point $x' \in \Delta(x)$ with $h(x') = 1$. Importantly, the fact that some true positives might need to put in effort to improve in order to be classified as positive does not count as a classification error in learning with improvements.

**Remark 2.1.** *Our improvement set based abstraction has a one-to-one correspondence with utility-maximizing agents, including the adversarial tiebreaking behavior, as follows. Set the agent utility to be $1$ at positive points $x$ with $h(x) = 1$, $0$ at negative points $x$ with $h(x) = 0$, and the utility of movement from $x$ to a point $x' \in \Delta(x)$ as a real number in $[0, 1]$. As an example, we could set the utility of movement to $\mathbf{1}[x \in \Delta(x)] \cdot h(x') \cdot \frac{2f^*(x')}{3}$. For this utility function, the agent has no incentive to move if either $\mathbf{1}[x \notin \Delta(x)]$ or $h(x') = 0$, else it has a utility of $\frac{2}{3}$ for ground-truth*

*negative points $f^*(x') = 0$ and $\frac{1}{3}$ for ground-truth positive points, so the agent moves in either case but prefers the former. Note that $\Delta(x)$ consists of the points to which the agent $x$ would ever consider (or is able to) move to, and its incentives to move to a point $x$ within $\Delta(x)$ are governed by $h(x')$ and $f^*(x')$ as described above.*

*Conversely, our improvement set based abstraction can be used to model an arbitrary utility function as follows. Without loss of generality say that the utility of being positive is $1$ and negative is $0$. Movements are associated with cost functions (negative utilities) for each $x \rightsquigarrow x'$ move, and costs outside of $(0, 1)$ can be ignored as the agent either always moves or never moves, irrespective of the classifier. We can define the improvement set $\Delta(x)$ to be the set of points where the cost is in $(0, 1)$. Now the agent with $h(x) = 0$ will move to a point $x'$ with $h(x) = 1$ as long as $x' \in \Delta(x)$, as their net utility is $1 - cost(x, x') > 0$. We can further incorporate worst-case tiebreaking by defining $\Delta(x)$ to only consist of points with $f^*(x') = 0$.*

*PAC-learning with improvements* is defined similarly to standard PAC-learning but for the above loss that incorporates agents' movements. A learning algorithm $\mathcal{A}$ has access to a training set consisting of $m$ samples $S \in \mathcal{X}^m$ drawn i.i.d. from a fixed but unknown distribution $\mathcal{D}$ over $\mathcal{X}$ and labeled by the ground truth $f^*$. The learner's population loss is $\text{LOSS}_{\mathcal{D}}(h, f^*) = \mathbb{P}_{x \sim \mathcal{D}}[\text{LOSS}(x; h, f^*)]$. Most of the results of this paper focus on the *realizable* setting in which $f^* \in \mathcal{H}$ and for which we can define PAC-learnability as follows:

**Definition 2.2.** *[5, Definition 2.2] A learning algorithm $\mathcal{A}$ PAC-learns with improvements a concept class $\mathcal{H}$ with respect to improvement function $\Delta$ and data distribution $\mathcal{D}$ with sample complexity $m := m(\varepsilon, \delta; \Delta, \mathcal{H}, \mathcal{D})$ if for any $f^* \in \mathcal{H}$ and $\varepsilon, \delta > 0$, the following holds: with probability at least $1 - \delta$, $\mathcal{A}$ takes in as input a sample $S \overset{i.i.d.}{\sim} \mathcal{D}^m$ labeled according to $f^*$ and outputs $h : \mathcal{X} \to \{0, 1\}$ such that $\text{LOSS}_{\mathcal{D}}(h, f^*) \leq \varepsilon$.*

## 3  Characterizing proper PAC-learning with improvements for any improvement function

In this section we prove a complete characterization of which concept classes are properly PAC-learnable with improvements for any improvement function $\Delta$. Our main conceptual advance is establishing a connection between PAC-learnability with improvements and the *minimally consistent* property that characterizes PAC-learnability with one-sided error [48, Chapter 2.4]. For a discussion on the previous results on proper PAC-learning with improvements which our theorem generalizes, see Appendix B. We first review relevant background on learning with one-sided error.

**Definition 3.1.** *An algorithm $A$ learns a concept class $\mathcal{H}$ with* one-sided error *if $A$ PAC-learns $\mathcal{H}$ and the concept output by $A$ does not have false positives.*

For a concept $f$, denote by $\text{graph}(f)$ the set of all examples for $f$, namely $\{(x, f(x))\}_{x \in \mathcal{X}}$. We say that $f$ is *consistent* with a set of examples $S \subset \mathcal{X} \times \mathcal{Y}$ if $S \subset \text{graph}(f)$. Let $S \subset \text{graph}(f)$ for some $f \in \mathcal{H}$. A concept $g \in \mathcal{H}$ is the *least $g \in \mathcal{H}$ consistent with $S$* if $g$ is consistent with $S$ and for all $h \in \mathcal{H}$ consistent with $S$, every instance classified positive by $g$ is also classified positive by $h$. Note that a least $g$ may not exist. A concept class $\mathcal{H}$ is *minimally consistent* if for each $f \in \mathcal{H}$ and each nonempty and finite subset $S \subset \text{graph}(f)$, there exists a least $g \in \mathcal{H}$ consistent with $S$. It is a textbook result that *minimally consistent* characterizes learning with one-sided error.

**Theorem 3.2.** *[48, Chapter 2.4] A concept class is PAC-learnable with one-sided error if and only if it is minimally consistent.*

To characterize PAC-learnability with improvements, we subtly modify the minimally consistent property to exclude sets of examples that are all labeled positively. We only require $\mathcal{H}$ to be minimally consistent on subsets $S$ that have at least one negatively labeled example, calling this asymmetric variant *nearly minimally consistent*:

**Definition 3.3.** *A concept class $\mathcal{H}$ is* nearly minimally consistent *if for each $f \in \mathcal{H}$ and each nonempty and finite subset $S \subset \text{graph}(f)$ that contains at least one negative example $(x, 0) \in S$, there exists a least $g \in \mathcal{H}$ consistent with $S$.*

Our main result is that the new *nearly minimally consistent* property we define above exactly characterizes PAC-learnability with improvements:

---

**Algorithm 1:** Proper learning with improvements

---

**Input**: Set $S$ consisting of $m$ i.i.d. sampled instances $(x_1, y_1), \ldots, (x_m, y_m)$
1:  **if** there exists $i$ such that $y_i = 0$ **then**
2:      **Output**: Least concept in $\mathcal{H}$ consistent with $S$
3:  **else**
4:      **Output**: Any concept in $\mathcal{H}$ consistent with $S$
5:  **end if**

---

**Theorem 3.4.** *A concept class $\mathcal{H}$ is properly PAC-learnable with improvements for all improvement functions $\Delta$ and all data distributions $\mathcal{D}$ with sample complexity $m = O\left(\frac{1}{\varepsilon}\left(d_{\mathrm{VC}}(\mathcal{H}) + \log \frac{1}{\delta}\right)\right)$ if and only if $\mathcal{H}$ has finite VC-dimension and is nearly minimally consistent.*

*Proof sketch.* We discuss the conceptual ideas needed to prove Theorem 3.4, relegating the full proof to Appendix B. To learn with one-sided error, we output *conservative* classifier, namely the minimal concept that is consistent with the training set $S$, because we cannot afford *any* false positives. In learning with improvements however, we can have false positives so long as points in $S$ do not "improve" to them. Assuming a worst-case improvement function $\Delta$, the only case where points in $S$ do not move to any false positive is when they were originally positively labeled. In other words, false positives are allowed only if $S$ consists of only positive labeled examples. It turns out that learning with improvements on such $S$ is easy: if we see only positive examples in the training set, with high probability the target concept $f^*$ originally consisted almost entirely of positive labels under $\mathcal{D}$. In this case we can simply output *any* concept $h$ consistent with $S$ to achieve low error. Our learning rule is given in Algorithm 1. □

Our theorem for proper PAC-learning with improvements in the realizable setting and its relation to the previous results from Attias et al. [5] as well as to learning with one-sided error can be neatly summarized using a Venn diagram (Fig. 1). The following two examples show that each of the Venn inclusions is strict.

**Example 3.5** (Separation between intersection closed and minimally consistent). *Suppose $\mathcal{X} = \{x_1, x_2\}$ and $\mathcal{H} = \{h_1, h_2\}$ with $h_1(x_1) = 1, h_1(x_2) = 0$ and $h_2(x_1) = 0, h_2(x_2) = 1$. Clearly $h_1 \cap h_2 \notin \mathcal{H}$ so $\mathcal{H}$ is not intersection-closed. Since knowledge of a single label tells us the target concept, there is at most one concept consistent with any non-empty set, so $\mathcal{H}$ is minimally consistent.*

**Example 3.6** (Separation between minimally consistent and nearly minimally consistent). *Let $\mathcal{X} = \{x_1, x_2, x_3\}$ and consider a concept class $\mathcal{H} = \{h_1, h_2, h_3\}$, where $h_i(x_j) = \mathbf{1}\left[i \neq j\right]$ for all $i, j \in [3]$. Note that $\mathcal{H}$ is not minimally consistent since there is no least hypothesis $g \in \mathcal{H}$ consistent with $S = \{(x_1, 1)\}$. However, one can show that $\mathcal{H}$ is nearly minimally consistent, the idea being that these singleton sets $S$ only contain positive examples and hence it is not required for a least $g$ consistent with $S$ to exist. One can also show that $\mathcal{H}$ is PAC-learnable with improvements.*

Importantly, our results establish that PAC-learnability with one-sided error implies proper PAC-learnability with improvements. Nonetheless, since the difference between the learning algorithms for these two settings lies only in what happens when the training set $S$ has only positive labels, learning with improvements is only *slightly* easier than learning with one-sided error.

**Comparison with strategic classification.** A natural question to ask is when a concept class is learnable for any $\Delta$ in the strategic classification model, where $\Delta(x)$ denotes the set of points $x'$ that $x$ can misreport. Recall that in the strategic classification, since $x$ can misreport as but not truly improve to any $x' \in \Delta(x)$, the loss function is the following:

$$\mathrm{Loss}(x; h, f^*) = \max_{x' \in \Delta_h(x)} \mathbf{1}\left[h\left(x'\right) \neq f^*\left(x\right)\right].$$

We claim that for any concept class in which there exists a concept such that greater than $\varepsilon$ fraction of the points have negative label and greater than $\varepsilon$ fraction have positive label, there is no algorithm that can PAC-learn for every $\Delta$. In order to get error at most $\varepsilon$, the learning algorithm $\mathcal{A}$ must output a hypothesis $h$ that labels at least one point positive, say $h(x_+) = 1$. However, since greater than than $\varepsilon$ fraction of the points have negative label, if we take $\Delta(x) = \mathcal{X}$ for all $x \in \mathcal{X}$ then all negative points can misreport as $x_+$, thus incurring strategic loss greater than $\varepsilon$. We conclude that

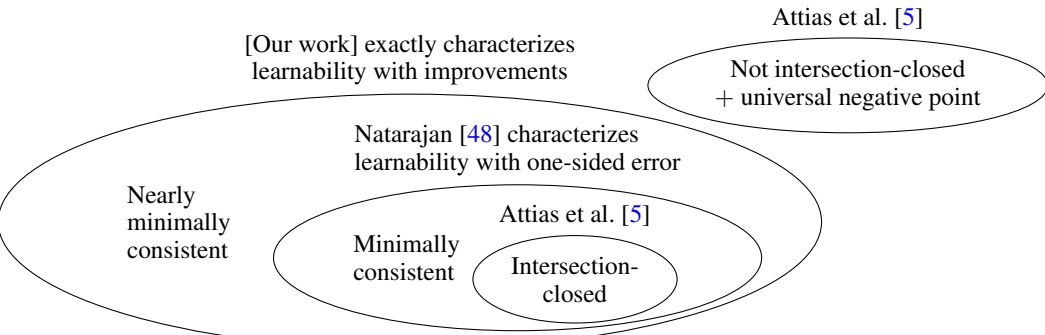

**Figure 1:** Relation between property PAC-learnability with improvements, PAC-learnability with one-sided error, and the nearly minimally consistent, minimally consistent, and intersection-closed properties, assuming that the concept class has finite VC-dimension.

PAC-learning for every $\Delta$ is impossible in the strategic setting unless all concepts either label all points in $\mathcal{X}$ negative or all points in $\mathcal{X}$ positive.

## 4 Improper PAC-learning with ball improvement sets

In this section we consider improper learning with improvements. We first note that in order to get interesting results, we must make some assumption on the improvement function $\Delta$. This is because we show that any concept class that is improperly PAC-learnable for *all* $\Delta$ must also be properly PAC-learnable, and we have already characterized proper PAC-learnability in Section 3 using the nearly minimally consistent property. (Recall that proper learnability trivially implies improper learnability.) Indeed, if $\Delta(x) = \mathcal{X}$ for all $x \in \mathcal{X}$ then the learning algorithm cannot make any false positives or else every negative instance can "improve" to such a false positive and incur a classification error. On the other hand, if $\Delta(x) = \{x\}$ for all $x \in \mathcal{X}$ then the learning algorithm cannot incur too many false negatives as any false negative is unable to improve. These two examples show that if we make no assumption on $\Delta$, the concept class must be learnable with one-sided error, which implies that it is *properly* PAC-learnable by our results in Section 3.

Therefore, in this section we will focus on PAC-learning of geometric concepts where $\mathcal{X} \subset \mathbb{R}^d$ for some $d$ and assume that the improvement function is $\Delta(x) = \{x' \in X \mid \|x' - x\|_2 \le r\}$, a Euclidean ball with radius $r$. (Our learnability upper bounds hold more generally for any improvement function for which $\Delta(x)$ *contains* a ball of radius $r$ centered at $x$.) The $\ell_2$-ball improvement set is well-studied in the strategic classification literature [55] and corresponds to the agent being able to misreport each of their features by a continuous and bounded amount. In Appendix C we construct a concept class with finite VC-dimension that shows that *proper* learning under $\ell_2$-ball improvement sets is intractable in general, which motivates why we consider *improper* learning. In stark contrast to proper learning, in the improper setting we show that under a generative data assumption known as *coverability* that *all* concept classes are PAC-learnable, and furthermore such distributions are learnable with the simple memorization learning rule (Algorithm 4).

**Coverability assssumption.**    To state our main result, we first describe the coverability assumption, which was introduced by Balcan et al. [14] in the context of robust learning in the presence of small, adversarial movements in the feature space.

**Definition 4.1.** *[14, Definition 1] A distribution $\mathcal{D}$ is $(\varepsilon, \beta, N)$-coverable if at least a $1 - \varepsilon$ fraction of probability mass of the marginal distribution $\mathcal{D}_{\mathcal{X}}$ can be covered by $N$ balls $B_1, \ldots, B_N$ in $\mathbb{R}^d$ of radius $\frac{r}{2}$ and of mass $\mathbb{P}_{\mathcal{D}_X}[B_k] \ge \beta$.*

Definition 4.1 means that one can cover most of the probability mass of the distribution with not too many balls. Coverability is important in many learning settings because it implies the following nice property: with enough samples from $\mathcal{D}$, with high probability it will be the case that all but $\varepsilon$ fraction of instances according to the marginal distribution $\mathcal{D}_{\mathcal{X}}$ will be distance at most $r$ from some sampled instance. In Appendix C we formalize this observation and also discuss its relation to the well-known *doubling dimension* of a distribution [22, 32].

**Sample complexity upper bound.** Our main positive result, proven in Appendix C, is that the memorization learning rule (Algorithm 4) can PAC-learn any distribution $\mathcal{D}$ for which the conditional distribution of positive instances are coverable:

**Theorem 4.2.** *Let the fraction of positive labels according to $\mathcal{D}$ be a constant independent of $\varepsilon$ and assume that $\mathcal{D}_{\mathcal{X}}^{+}$, the conditional marginal distribution for positive instances, is $(\varepsilon, \beta, N)$-coverable. Then with probability at least $1 - \gamma$, a predictor $h$ learned using Algorithm 4 from $m = O\left(\frac{1}{\beta} \log \frac{N}{\gamma}\right)$ i.i.d. samples from $\mathcal{D}$ has improvement loss $\mathrm{Loss}_{\mathcal{D}}(h, f^*) \leq \varepsilon$.*

We remark that the the parameter $\beta$, which is the probability mass of each ball, implicitly depends on the improvement radius $r$. For example, if the data distribution is uniform over a bounded set in $\mathbb{R}^d$, then a ball of radius $r$ has probability mass proportional to $r^d$. Interestingly, Theorem 4.2 requires only an assumption on the marginal distribution $\mathcal{D}_{\mathcal{X}}$ and is agnostic to the labels and the concept class. In particular, it is possible to improperly learn concept classes with infinite VC dimension so long as Definition 4.1 is satisfied. Furthermore, it is even possible to achieve *zero error* with high probability in many natural situations, which reinforces the conceptual advance that learning with improvements often yields lower generalization error than standard PAC-learning. For example, Proposition C.4 implies that if $\mathcal{X} \subset \mathbb{R}^d$ has a finite diameter then $\mathcal{X}$ is coverable with $\varepsilon = 0$, resulting in zero error with high probability by Theorem 4.2.

**Sample complexity lower bound.** Even though the memorization learning rule is very simple, it is the best one can hope for under the coverability assumption. Formally, in Appendix C we prove a lower bound for the number of samples needed to improperly PAC-learn with improvements:

**Theorem 4.3.** *There exists a concept class $\mathcal{H}$ and a $\left(0, \beta, \frac{1}{\beta}\right)$-coverable data distribution $\mathcal{D}$ such that $\Omega\left(\frac{1}{\beta} \log \frac{1}{\beta}\right)$ samples are necessary to achieve $\beta$ improvement loss with high probability.*

Note that our upper bound (Theorem 4.2) can be generalized to any improvement function where $\Delta(x) \supset \left\{x' \in \mathbb{R}^d \mid \|x' - x\|_2 \leq r\right\}$, namely the improvement region for any $x$ contains a ball of radius $r$. Additionally, our lower bound (Theorem 4.3) can be generalized to any improvement function where $\Delta(x) \subset \left\{x' \in \mathbb{R}^d \mid \|x' - x\|_2 \leq r\right\}$, namely the improvement region for any $x$ is contained a ball of radius $r$. These two observations imply that our results are robust to the exact shape of the improvement set. Since we should think of $N$ in Theorem 4.2 of order $\Theta\left(\frac{1}{\beta}\right)$, which corresponds to $N$ balls each of mass $\beta$ covering all but $\varepsilon$ fraction of the instance space $\mathcal{X}$, Theorem 4.3 shows that Theorem 4.2 is tight up to logarithmic factors in $\beta$. This implies that the very simple memorization rule learns with improvements with a near-optimal sample complexity in the improper setting under a coverability assumption.

**Comparison with strategic classification.** One interesting observation is that the memorization learning rule outputs a predictor that has monotonically decreasing improvement loss as the improvement region enlarges. This is because there are no false positives, and if false positives improve they must do so to a point with ground-truth positive label. A larger improvement set gives false positives weakly more points to improve to and hence can only reduce error. On the other hand, a larger movement set in strategic classification is generally expected to increase error, as this gives more chances for a negative point to mimic a positive point. For example, in an extreme setting where $\mathcal{X}$ is bounded, the movement radius $r$ is large enough to cover the entirety of $\mathcal{X}$, and there is at least one point with positive label, then *every* negative point incurs classification error.

## 5 Learning linear separators optimally under bounded noise

As mentioned in the introduction, there is a large and growing literature that studies PAC-learning with different types of label noise [10]: random classification noise [23, 18, 26], Massart noise [7, 8, 31, 25, 28], malicious noise Kearns and Li [39], Klivans et al. [41], Awasthi et al. [6, 9], and nasty noise [21, 30, 12]. Our understanding of learning with noise in the presence of strategic or improving agents is far more limited. Braverman and Garg [19] study learning in the presence of strategic agents with feature noise but no label noise. In contrast, we focus here on label noise. We develop optimal algorithms for learning linear separators in the improvements setting with bounded label noise under isotropic log-concave distributions.

Concretely, as before we have a distribution $\mathcal{D}$ over the points in $\mathcal{X}$. However, the learner does not see perfect labels $f^*(x)$ in the training sample. Instead, the labels $y \in \{0, 1\}$ are given by a noisy label distribution $y \mid x \sim \mathcal{N}$. An example is when $y$ comes from crowdsourced data, where one typically assumes that for any given $x$ the majority of labelers, but not all of them, will label the point correctly. Formally, $f^*_{\text{bayes}}(x) = \text{sign}(\mathbb{E}[y \mid x] - \frac{1}{2})$. The *bounded* or *Massart noise* model has a parameter $\nu < \frac{1}{2}$ corresponding to an upper bound on the noise of any point $x$, i.e. $\nu(x) = \mathbb{E}_{\mathcal{N}}[\mathbf{1}[y \neq f^*_{\text{bayes}}(x)]) \mid x] \leq \nu$. A special case is the *random classification noise* (RCN), where all the labels of all points are flipped by equal probability $\nu < \frac{1}{2}$, or $\mathbb{E}_{\mathcal{N}}[\mathbf{1}[y \neq f^*_{\text{bayes}}(x)]) \mid x] = \nu$.

In the presence of noise, the goal is to minimize the expected loss which modifies the loss (Eq. (2.2)) in the noiseless case by considering movement of the agent $x$ to the point in the reaction set $\Delta_h(x)$ which has the largest expected loss (expectation is over the noise as below):

$$\text{Loss}_{\mathcal{N}}(x; h) = \max_{x' \in \Delta_h(x)} \mathbb{E}_{y' \mid x' \sim \mathcal{N}} [h(x') \neq y'].$$

**General reduction to PAC-learning with noise.** In Appendix D we show a general reduction from learning linear separators with noise in the improvements settings to learning with noise in the standard PAC-setting:

**Theorem 5.1.** *Suppose the improvement region for each point $x \in \mathcal{X} = \mathbb{R}^d$ is given by $\Delta(x) = \{x' \mid \arccos(\langle x, x' \rangle) \leq r\}$ and $\mathcal{H}$ is the class of homogeneous linear separators $\mathcal{H} = \{x \mapsto \text{sign}(w^T x) : w \in \mathbb{R}^d\}$. Let $\mathcal{D}$ be an isotropic log-concave distribution over the instance space $\mathcal{X}$, realizable by some $f^*_{bayes} \in \mathcal{H}$. Suppose the learner has access to noisy labels with bounded noise $\nu(x) = \mathbb{E}_{\mathcal{N}}[\mathbf{1}[y \neq f^*_{bayes}(x)]) \mid x] \leq \nu$. Finally, suppose that for any $\varepsilon, \delta \in (0, \frac{1}{2})$ given access to a noisy sample of size $m = p(\frac{1}{\varepsilon}, \log \frac{1}{\delta})$, there is a proper learner in the standard PAC-setting that outputs $\hat{h}$ with error at most $\nu + \varepsilon$, with probability $1 - \delta$ over the draw of the sample. For any $\delta \in (0, \frac{1}{2})$, a noisy sample of size $m = p\left(O\left(\frac{1}{(1-2\nu)r}\right), \log \frac{1}{\delta}\right)$ is sufficient to PAC-learn with improvements an improper classifier $\hat{f}$ such that for any $x \in \mathcal{X}$, $\text{Loss}_{\mathcal{N}}(x; \hat{f}) = \text{Pr}_{\mathcal{N}}[y \neq f^*_{bayes}(x)]$ with probability at least $1 - \delta$. That is, our classifier is Bayes optimal.*

*Proof sketch.* Our overall idea is as follows. Suppose there is proper learning algorithm that attains low error rate given sufficiently many samples from the distribution. For nice data distributions (e.g. for uniform, Gaussian, or log-concave distributions), in the case of linear separators, small error implies a small angle from the target concept $f^*_{\text{bayes}}$ [43, 10]. Our approach is to run these algorithms on sufficiently many samples such that the angle $\theta$ between $f^*_{\text{bayes}}$ and the learned $\hat{f}$ is small relative to the agent movement budget $r$. Then we use $\hat{f}$ to construct a conservative (improper) classifier $\tilde{f}$ such that all agents between $f^*_{\text{bayes}}$ and $\tilde{f}$ are able to move to get positively classified according to $\tilde{f}$, but there is no point classified negative by $f^*_{\text{bayes}}$ but positive by $\tilde{f}$. $\qquad \square$

**Instantiation for well-studied noise classes.** Theorem 5.1 implies sample complexity bounds for Bayes optimal learning with improvements in the presence of bounded noise by simply plugging in the corresponding sample complexity bounds from the standard PAC learning setting. For instance, for RCN the sample complexity is well-known to be $\tilde{O}\left(\frac{d}{\varepsilon}\right)$ [10], implying $\tilde{O}\left(\frac{d}{r(1-2\nu)}\right)$ for optimal learning with improvements. Similarly, for Massart noise [44], Theorem 5.1 implies a $\tilde{O}\left(\frac{d}{r^2(1-2\nu)^2}\right)$ sample complexity bound. Note that while prior work on standard PAC learning only achieves $\text{OPT} + \varepsilon$, we achieve exactly $\text{OPT}$, the error of $f^*_{\text{bayes}}(x)$.

**Discussion.** To the best of our knowledge we are the first to study and design classifiers for *label* noise when agents react to the classifier. As in Section 4, in the strategic improvements model we can achieve *smaller* error than standard PAC-learning for learning linear separators with noise, in this case reaching the Bayes optimal error. Braverman and Garg [19] study designing classifiers under *feature* noise. One surprising result of their work is that it is sometimes possible for the classifier to achieve *higher* accuracy when the agents' features are noisy under strategic classification. This is counterintuitive because noisy features cannot help when there is no strategic manipulation. It is an interesting open question whether feature noise can reduce error under improvements as well.

---
**Algorithm 2:** Realizable online learning: Risk-averse majority vote
---
**Input**: Concept class $\mathcal{H}$, maximum degree of the graph $\Delta_G$.

1: Initialize $H \leftarrow \mathcal{H}$.
2: **for** $t = 1, 2, \ldots$ **do**
3:   For each node $x$, set

$$\hat{h}^{(t)}(x) = \begin{cases} 1, & \text{if } |\{h \in H \mid h(x) = 1\}| \geq \frac{\Delta_G}{\Delta_G+1}|H|, \\ 0, & \text{otherwise.} \end{cases}$$

4:   $\Delta^+ \leftarrow \{x \in \Delta(x^{(t)}) \mid \hat{h}^{(t)}(x) = 1\}$.
5:   $H' \leftarrow \{h \in H \mid h(x') = 1 \text{ for all } x' \in \Delta^+\}$.
6:   **if** there is a mistake, and $\hat{h}^{(t)}(x^{(t)}) = 0$ **then**
7:      If $|\Delta^+| = 0$, $H \leftarrow \{h \in H \mid h(x^{(t)}) = 1\}$.
8:      Else, $H \leftarrow H \setminus H'$.
9:   **end if**
10:   If there is a mistake and $\hat{h}^{(t)}(x^{(t)}) = 1$, $H \leftarrow \{h \in H \mid h(x^{(t)}) = 0\}$.
11: **end for**
---

## 6 Online learning on a graph

Attias et al. [5] study the sample complexity of learning with improvements in a general discrete graph model in the statistical learning setting. Here we will study mistake bounds in the natural online learning version of their model. The nodes of the graph correspond to agents (points) and the (undirected, unweighted) edges govern the improvement function, i.e. the agents can move to their neighboring nodes in order to potentially improve their classification. Formally, let $G = (V, E)$ denote an undirected graph. The vertex set $V = \{x_1, \ldots, x_n\}$ represents a fixed collection of $n$ points corresponding to a finite instance space $\mathcal{X}$. The edge set $E \subseteq V \times V$ captures the adjacency information relevant for defining the improvement function. More precisely, for a given vertex $x \in V$, the improvement set of $x$ is given by its neighborhood in the graph, i.e. $\Delta(x) = \{x' \in V \mid (x, x') \in E\}$. Let $f^* : V \to \{0, 1\}$ represent the target labeling (partition) of the vertices in the graph $G$.

In the online setting, for each round $t = 1, 2, \ldots$, the learner sees a node $x^{(t)} \in V$ and must make a prediction $h^{(t)}$ for all nodes. The true label $f^*(x^{(t)})$ is then revealed and the learner is said to suffer a *mistake* if there there is some $x' \in \Delta(x^{(t)})$ such that $h^{(t)}(x') \neq f^*(x')$. The learner only knows whether a mistake was made, without learning about $x'$. The goal of the learner is to minimize the total number of mistakes across all rounds. Our main contributions are new algorithms for online learning with improvements in both the realizable and agnostic settings. All proofs of theorems in this section are in Appendix E.

**Realizable online learning.**   In the *realizable* setting, $f^* \in \mathcal{H}$, the concept space consists of valid labelings that the learner is allowed to output. In the standard learning setting (where agents cannot improve), the majority vote algorithm (which predicts using the majority label of consistent classifiers in $\mathcal{H}$, and discards all the inconsistent classifiers at every mistake) achieves a mistake upper bound of $\log |\mathcal{H}|$. We first construct an example (Example E.1) where the majority vote algorithm can result in an unbounded number of mistakes in the learning with improvements setting.

A risk-averse modification of the majority vote algorithm, requiring a certain super-majority for positive classification, can avoid the unbounded mistakes by the standard majority vote algorithm. This is in stark contrast to the online learning algorithm of Ahmadi et al. [3] for the strategic classification setting, where a super-majority is used for negative classification. Another modification from standard majority vote (see Algorithm 2) is a change in the way classifiers are discarded when a mistake is made, taking the predictions on the neighboring nodes into account.

**Theorem 6.1.** *Algorithm 2 makes at most $(\Delta_G + 1) \log |\mathcal{H}|$ mistakes, where $\Delta_G$ is the maximum degree of a vertex in $G$.*

---

**Algorithm 3:** Agnostic online learning: Risk-averse weighted majority vote

---

**Input**: Concept class $\mathcal{H}$, maximum degree of the graph $\Delta_G$.

1: Initialize $w_h = 1$ for each $h \in \mathcal{H}$.
2: **for** $t = 1, 2, \dots$ **do**
3:     $W_t \leftarrow \sum_{h \in \mathcal{H}} w_h$
4:     For each node $x$, set
5:     $W_t^+ \leftarrow \sum_{h \in \mathcal{H} | h(x) = 1} w_h$

$$\hat{h}^{(t)}(x) = \begin{cases} 1, & \text{if } W_t^+ \geq \frac{\Delta_G}{\Delta_G + 1} W_t, \\ 0, & \text{otherwise.} \end{cases}$$

6:     $\Delta^+ \leftarrow \{x \in \Delta(x^{(t)}) \mid \hat{h}^{(t)}(x) = 1\}$.
7:     $H' \leftarrow \{h \in \mathcal{H} \mid h(x') = 1 \text{ for all } x' \in \Delta^+\}$.
8:     **if** there is a mistake, and $\hat{h}^{(t)}(x^{(t)}) = 0$ **then**
9:        If $|\Delta^+| = 0$, $H_t^- \leftarrow \{h \in H \mid h(x^{(t)}) = 0\}$, $w_h \leftarrow w_h / 2$ for each $h \in H_t^-$.
10:       Else, $w_h \leftarrow w_h / 2$ for each $h \in H'$.
11:     **end if**
12:     If there is a mistake and $\hat{h}^{(t)}(x^{(t)}) = 1$, $H_t^+ \leftarrow \{h \in H \mid h(x^{(t)}) = 1\}$, $w_h \leftarrow w_h / 2$ for each $h \in H_t^+$.
13: **end for**

---

**Agnostic online learning.** In the agnostic setting, we remove the realizability assumption that there exists a perfect classifier $f^* \in \mathcal{H}$. Instead, we will try to compete with smallest number of mistakes achieved by any classifier in $\mathcal{H}$, denoting this by $\mathsf{OPT}$. Our online learning algorithm will be a risk-averse version of the *weighted majority vote* algorithm.

We maintain a set of weights $\{w_h \mid h \in \mathcal{H}\}$ for each concept in the concept space. Initially, $w_h = 1$ for each concept $h \in \mathcal{H}$. Let $W_t^+$ denote the sum of weights of experts that predict a node $x$ as positive in round $t$, and $W = \sum_h w_h$ denote the total weight. Then the risk-averse online learner predicts $x$ as positive if $W_t^+ \geq \frac{\Delta_G}{\Delta_G + 1} W$, and negative otherwise. Finally, if there is a mistake, then we halve the weights of certain classifiers (as opposed to discarding them in Algorithm 2).

**Theorem 6.2.** *Let $G$ be any graph with maximum degree $\Delta_G \geq 1$. Algorithm 3 makes $O(2(\Delta_G + 1)(\mathsf{OPT} + \log |\mathcal{H}|))$ mistakes.*

**Mistake lower bounds.** Finally, we show lower bounds on the number of mistakes made by any *deterministic* learner against an adaptive adversary in both the realizable and agnostic settings.

**Theorem 6.3.** *For any $\Delta > 1$, there exist a graph $G$ with any maximum degree $\Delta$, a hypothesis class $\mathcal{H}$, and an adaptive adversary such that any deterministic learning algorithm makes at least $\Delta \cdot \mathsf{OPT}$ mistakes in the agnostic setting and $\Delta - 1$ mistakes in the realizable setting.*

The lower bound in Theorem 6.3 is against deterministic algorithms. If we allow for the use of randomness, it may be possible to remove the factor of $\Delta_G$ in the statement of Theorem 6.2 by using a modified version of the Hedge algorithm similar to [3, Algorithm 3]. We leave this as an open question.

## 7 Conclusion

In this paper we study statistical learning under strategic improvements. We develop new algorithms for proper learning with any improvement function, improper learning with Euclidean ball improvement sets, learning with noise, and online learning. Our work opens up several exciting new future directions. For example, it would be interesting to extend the learning with improvements model beyond the binary classification setting to multi-label and regression settings. This would have practical applications to any learning problem that involves several classes such as assigning credit scores or deciding loan amounts. Also, we can study learnability under label noise in strategic classification, as well as under feature noise in our improvements model.

## Acknowledgments

This work was supported in part by the National Science Foundation under grants ECCS-2216899, ECCS-2216970, and by a NSF Graduate Research Fellowship.

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

## A  Discussion of related work

Our paper is related to several works studying learning in strategic and adversarial environments:

- **Strategic classification.** Taking into account how utility-maximizing agents can strategically "game" the classifier is an important research area in societal machine learning [37, 46, 19, 1, 35]. The first papers in this area modeled strategic gaming behavior as a Stackelberg game [36, 20] where negatively classified agents manipulate their features to obtain more favorable outcomes if the benefits outweigh the costs.

- **Learning with improvements.** Kleinberg and Raghavan [40] are the first to consider *learning with improvements*, which is when agents manipulate but also genuinely improve their features. Ahmadi et al. [2] consider a similar improvement setting and propose classification models that balance maximizing true positives with minimizing false positives. Prior work has studied the inherent learnability of concepts in the strategic manipulation setting [54, 27, 42] but not in the strategic improvement setting. Attias et al. [5] propose to study the statistical learnability of concept classes, the sample complexity of learning, and the ability to achieve zero-error classification in the improvement setting. Haghtalab et al. [35] also study the sample complexity of learning in the presence of improving agents, but they optimize for social welfare by maximizing the fraction of true positives after improvement and primarily focus on linear mechanisms. In contrast, we focus on *classification error* in which false positives matter, leading to fundamentally different properties of good classifiers.

- **Reliable learning.** Learning with improvements is also related to reliable machine learning [49, 33] in which learner may abstain from classification to avoid mistakes. The goal in reliable learning is to tradeoff *coverage*, the fraction of classified points, against classification error. The conservative classification paradigm that serves as a basis for many of our algorithms also has a similar flavor to learning with one-sided error [47, 38] in which the learned classifier is not allowed to have any false positives. There are connections between strategic classification and adversarial learning [54], but it remains an interesting open question if similar connections can be established between learning with improvements and adversarial learning [13, 15, 17].

- **Learning with noise.** There is a large and growing literature that studies PAC-learning with different types of label noise [10]: random classification noise [23, 18, 26], Massart noise [7, 8, 31, 25, 28], malicious noise Kearns and Li [39], Klivans et al. [41], Awasthi et al. [6], and nasty noise [21, 30, 12]. Recent work [29, 24] has developed optimal approaches for learning margin halfspaces with bounded label noise. Braverman and Garg [19] study learning in the presence of strategic agents with feature noise but no label noise. The problem of learning with noise in the presence of strategic agents is understudied and a very relevant direction for future work.

- **Online strategic classification.** Ahmadi et al. [3, 4], Cohen et al. [27], Shao et al. [50] study the problem of *strategic* online binary classification. Our discrete graph model for online learning is similar to theirs, agents are nodes that can potentially move to their neighbors, except that we consider true movements that can change the agents' labels (as opposed to just their classification). This distinction leads to surprisingly different good online learners. Connections with adversarially robust online learning (e.g. Goldblum et al. [34], Balcan et al. [11], Sharma [51], Sharma and Suggala [52]) are less well understood, especially in multi-task and meta-learning settings.

## B  Characterizing proper PAC-learning with improvements for any improvement function

**Prior work.** Attias et al. [5] prove a *sufficient* condition for learnability based on a property called *intersection-closed*, which we define below.

**Definition B.1** (Closure operator of a set). *For any set $S \subseteq \mathcal{X}$ and any hypothesis class $\mathcal{H} \subseteq 2^{\mathcal{X}}$, the* closure *of $S$ with respect to $\mathcal{H}$, denoted by $\mathrm{CLOS}_{\mathcal{H}}(S) : 2^{\mathcal{X}} \to 2^{\mathcal{X}}$, is defined as the intersection of all hypotheses in $\mathcal{H}$ that contain $S$, that is, $\mathrm{CLOS}_{\mathcal{H}}(S) = \bigcap_{h \in \mathcal{H}, S \subseteq h} h$. In words, the closure of $S$ is the smallest hypotheses in $\mathcal{H}$ which contains $S$. If $\{h : \mathcal{H} : S \subseteq h\} = \emptyset$, then $\mathrm{CLOS}_{\mathcal{H}}(S) = \mathcal{X}$.*

**Definition B.2** (Intersection-closed classes). *A hypothesis class $\mathcal{H} \subset 2^{\mathcal{X}}$ is* intersection-closed *if for all finite $S \subseteq \mathcal{X}$, $\mathrm{CLOS}_{\mathcal{H}}(S) \in \mathcal{H}$. In words, the intersection of all hypotheses in $\mathcal{H}$ containing an arbitrary subset of the domain belongs to $\mathcal{H}$. For finite hypothesis classes, an equivalent definition states that for any $h_1, h_2 \in \mathcal{H}$, the intersection $h_1 \cap h_2$ is in $\mathcal{H}$ as well Natarajan [47].*

There are many natural intersection-closed concept classes, for example, axis-parallel $d$-dimensional hyperrectangles, intersections of halfspaces, $k$-CNF boolean functions, and linear subspaces.

**Theorem B.3.** *[5, Theorem 4.7] Let $\mathcal{H}$ be an intersection-closed concept class on instance space $\mathcal{X}$. There is a learner that PAC-learns with improvements $\mathcal{H}$ with respect to any improvement function $\Delta$ and any data distribution $\mathcal{D}$ given a sample of size $O\left(\frac{1}{\epsilon}(d_{VC}(\mathcal{H}) + \log\frac{1}{\delta})\right)$, where $d_{VC}(\mathcal{H})$ denotes the VC-dimension of $\mathcal{H}$.*

Attias et al. [5] also prove the following necessary condition for learnability:

**Theorem B.4.** *[5, Theorem 4.8] Let $\mathcal{H}$ be any concept class on a finite instance space $\mathcal{X}$ such that at least one point $x' \in \mathcal{X}$ is classified negative by all $h \in \mathcal{H}$ (i.e. $\{x \mid h(x) = 0 \text{ for all } h \in \mathcal{H}\} \neq \emptyset$), and suppose $\mathcal{H}\mid_{\mathcal{X}\setminus\{x'\}}$ is not intersection-closed on $\mathcal{X} \setminus \{x'\}$. Then there exists a data distribution $\mathcal{D}$ and an improvement function $\Delta$ such that no proper learner can PAC-learn with improvements $\mathcal{H}$ with respect to $\Delta$ and $\mathcal{D}$.*

Note that Theorem B.4 only applies when there is a point that is classified negative by all concepts in $\mathcal{H}$ and says nothing about learnability when this condition does not hold. Our main result, Theorem 3.4, shows an exact characterization of which concept classes are properly PAC-learnable with improvements for all improvement functions which generalizes both Theorem B.3 and Theorem B.4.

**Characterizing proper PAC-learnability with improvements.**   We prove Theorem 3.4.

*Proof of Theorem 3.4.* Note that if $\Delta(x) = \{x\}$ for all $x$ then learning with improvements reduces to vanilla PAC-learning, so finite VC-dimension is necessary by the Fundamental Theorem of Statistical Learning. Assuming now that $\mathcal{H}$ has finite VC-dimension, we first show that if a concept class $\mathcal{H}$ is *not* nearly minimally consistent then there exists an improvement function $\Delta$ and a data distribution $\mathcal{D}$ for which $\mathcal{H}$ is not PAC-learnable with improvements. Let $f \in \mathcal{H}$ and $S \subset \text{graph}(f)$ be such that $S$ contains a negative example $(x_-, 0) \in S$ and there is no least concept consistent with $S$. Let $\mathcal{D}$ be the uniform distribution over $S_{\mathcal{X}}$ and suppose that the improvement function is

$$\Delta(x_-) = \begin{cases} \mathcal{X} & \text{if } x = x_- \\ \emptyset & \text{otherwise.} \end{cases}$$

Suppose that a learning algorithm outputs a concept $g \in \mathcal{H}$. If $g$ is not consistent with $S$, then either $g(x_-) = 1$ or $g(x) = 0$ for some $x \in S, x \neq x_-$. Since points in $S$ other than $x_-$ cannot move, such points cannot improve and hence $g$ suffers constant error. Hence we can assume that $g$ is consistent with $S$. Since there is no least concept consistent with $S$, there exists a concept $h$ also consistent with $S$ and a point $x_+$ for which $g(x_+) = 1$ and $h(x_+) = 0$. Note that $h$ could have very well been the target concept since $\mathcal{D}$ is uniformly supported on $S$ for which both $g$ and $h$ are consistent with. However, note that $x_-$ can in the worst case "improve" to $x_+$ since $g(x_+) = 1$, but then outputting $g$ suffers constant error since the ground truth is $h(x_+) = 0$. We conclude that any learning algorithm must suffer constant error on $\mathcal{D}$ for the improvement function $\Delta$ defined above.

The other direction is to show that if a concept class $\mathcal{H}$ *is* nearly minimally consistent then $\mathcal{H}$ is PAC-learnable with improvements. The learning algorithm we use (Algorithm 1) is very similar to the one used to learn with one-sided error [48, Chapter 2.4]. However, we need to specify what concept the algorithm outputs when the training set consists of only positive examples, in which case there may not be a least concept consistent with the examples so far. In this case it turns outputting any concept consistent with the training set $S$ will work. Informally, this is because if $S$ consists of only positive examples, then with high probability it must have been the case that $f^*$ positively labels nearly all points according to $\mathcal{D}$. By the Fundamental Theorem of Statistical Learning, the concept $h_S$ output by the learning algorithm agrees with $f^*$ on nearly all of $\mathcal{D}$, which also means that $h_S$ positively labels nearly all points. Since positively label points do not move, then $h_S$ has low error in the improvement setting as well.

We now give the formal proof. For a number of samples $m = O\left(\frac{1}{\varepsilon}\left(d_{\text{VC}}(\mathcal{H}) + \log\frac{1}{\delta}\right)\right)$, we know, for example see [48, Theorem 2.1], that the concept $h_S$ output by the the learning algorithm satisfies $\mathbb{P}_{x\sim\mathcal{D}}\left[h_S(x) \neq f^*(x)\right] \leq \frac{\varepsilon}{2}$ with probability at least $1 - \frac{\delta}{2}$ for any target concept $f^* \in \mathcal{H}$. We split into two cases.

- **Case 1:** $\mathbb{P}_{x\in\mathcal{D}}\left[f^*(x) = 1\right] \geq 1 - \frac{\varepsilon}{2}$. Then the concept $h_S$ disagrees with $f^*$ on at most $\frac{\varepsilon}{2}$ fraction of points according to $\mathcal{D}$, so $\mathbb{P}_{x\in\mathcal{D}}[h_S(x) = f^*(x) = 1] \geq 1 - \varepsilon$. Such $x$

are positively classified so they do not move and hence incur zero improvement loss. We conclude that with probability at least $1 - \frac{\delta}{2}$ the improvement loss is at most $\varepsilon$.

- **Case 2:** $\mathbb{P}_{x \in \mathcal{D}}\left[f^*(x) = 1\right] \leq 1 - \frac{\varepsilon}{2}$. Then after $m$ samples, the probability that the training set $S$ consists of a negative example is at least $1 - \frac{\delta}{2}$. For any $x \in \mathcal{X}$ with $h_S(x) = 1$, $x$ does not move and hence the improvement loss is the same as the 0-1 loss $\mathbf{1}\left[h_S(x) \neq f^*(x)\right]$. For any $x \in \mathcal{X}$ with $h_S(x) = 0$, suppose that $x$ moves to a point $x'$, possibly equal to $x$. If $x' = x$ then the improvement loss is again the same as the 0-1 loss. Otherwise we can assume $x' \neq x$, which means that $h_S(x') = 1$ since $x$ only moves if it can improve. If $f^*(x) = 1$ then the loss $\mathbf{1}\left[h_S(x') \neq f^*(x')\right]$ is at most the 0-1 loss $\mathbf{1}\left[h_S(x) \neq f^*(x)\right] = 1$. Finally, we can assume $f^*(x) = 0$. Since $S$ consists of a negative example, $h_S$ is the least concept consistent with $S$, so $h_S(x') = 1 \implies f^*(x') = 1$ in which case $x$ has zero improvement loss. By the union bound, with probability at least $1 - \delta$ the improvement loss is at most $\frac{\varepsilon}{2}$.

In both cases we conclude that with probability at least $1 - \delta$ the improvement loss is at most $\varepsilon$. $\qquad\square$

## C  Improper PAC-learning with ball improvement sets

**Proper learning with ball improvement sets is intractable.** We construct a simple concept class that has finite VC-dimension but is impossible to properly PAC-learn under $\ell_2$-ball improvement sets.

**Example C.1.** *Consider a union of two intervals, which clearly has finite VC-dimension. Let the instance space $\mathcal{X}$ be $[0,1]$, let $\mathcal{H} = \left\{h_b : h_{abc}(x) = 1 \text{ iff } x \in [\frac{1}{4}, b) \bigcup (b, \frac{3}{4}]\right\}$, and let $\mathcal{D}$ be the uniform distribution over $[0,1]$. Let $r = \frac{1}{2}$ and consider a target function $f^* = h_b$ with $b$ chosen uniformly in $\left(\frac{1}{4}, \frac{3}{4}\right)$.*

*For any learning algorithm, with probability 1 the learner will not see the point $b$ in its training data, so it learns nothing from its training data about the location of $b$. Note that $\mathrm{Loss}(f^*, f^*) = 0$. On the other hand, we claim that for any other $h = h_b \in \mathcal{H}$ we have $\mathrm{Loss}(h_b, f^*) \geq \frac{1}{4}$. Without loss of generality assume $b \leq \frac{1}{2}$. Note that in the worst-case, all points $x \leq \frac{1}{4}$, which are negatively labeled, will move to $b$, incurring loss at least $\frac{1}{4}$.*

**Covering lemma.** We formally state and prove that with enough samples from $\mathcal{D}$, with high probability it will be the case that all but $\varepsilon$ fraction of instances according to the marginal distribution $\mathcal{D}_{\mathcal{X}}$ will be distance at most $r$ from some sampled instance.

**Lemma C.2.** *[14, Theorem 4.4] Suppose that $x_1, \ldots, x_m$ are $m$ instances i.i.d. sampled from the marginal distribution $\mathcal{D}_{\mathcal{X}}$. If $\mathcal{D}$ is $(\varepsilon, \beta, N)$-coverable, for sufficiently large $m = \Omega\left(\frac{1}{\beta}\log\frac{N}{\gamma}\right)$, with probability at least $1 - \gamma$ over the sampling we have $\mathbb{P}\left[\bigcup_{i=1}^m B(x_i, r)\right] \geq 1 - \varepsilon$.*

For completeness we provide a proof of the covering lemma from Balcan et al. [14].

*Proof of Lemma C.2.* Fix ball $B_i$ in the cover from Definition 4.1. Let $\overline{\mathcal{B}_i}$ denote the event that no point is drawn from ball $B_i$ over the $m$ samples. Since successive draws are independent and by definition $\mathbb{P}_{\mathcal{D}_{\mathcal{X}}}[\mathcal{B}_i] \geq \beta$, we have

$$\mathbb{P}\left[\overline{\mathcal{B}_i}\right] \leq (1 - \beta)^m \leq \exp(-\beta m).$$

By a union bound over $N$ balls we have

$$\mathbb{P}\left[\bigcup_i \overline{\mathcal{B}_i}\right] \leq N \cdot \exp(-\beta m) \leq \gamma$$

for $m \geq \frac{1}{\beta}\log\frac{N}{\gamma}$. Therefore, with probability at least $1 - \gamma$ we have

$$\bigcup_{i=1}^m B(x_i, r) \supset \bigcup_{k=1}^N B_k \implies \mathbb{P}\left[\bigcup_{i=1}^m B(x_i, r)\right] \geq \mathbb{P}\left[\bigcup_{k=1}^N B_k\right] \geq 1 - \varepsilon$$

since for all $k \in [N]$ there is a sample $x_{i_k} \in B_k$ and $B_k$ is a ball of radius $\frac{r}{2}$. $\qquad\square$

---

**Algorithm 4:** Memorization learning rule

---

**Input**: $m$ instances $(x_1, y_1), \ldots, (x_m, y_m)$
1: Initialize $h(x) = 0$ for all $x \in \mathbb{R}^d$
2: **for** $i = 1, \ldots, m$ **do**
3:    If $y_i = 1$, set $h(x_i) = 1$
4: **end for**
**Output**: $h$

---

**Doubling dimension.** The coverability property of a distribution is closely related to the well-known notion of *doubling dimension* of a metric space [22, 32].

**Definition C.3** (Doubling dimension). *A measure $\mathcal{D}_{\mathcal{X}}$ has* doubling dimension $d'$ *if for all points $x \in \mathcal{X}$ and all radii $r > 0$, we have $\mathcal{D}_{\mathcal{X}}(B(x, 2r)) \leq 2^{d'} \cdot \mathcal{D}_{\mathcal{X}}(B(x, r))$.*

Note that the uniform distribution on Euclidean space $\mathbb{R}^d$ has doubling dimension $d' = \Theta(d)$. Doubling dimension of a data distribution has been used to obtain sample complexities of generalization for learning problems [22] as well as to give bounds on cluster quality for nearest-neighbor based clustering algorithms in the distributed learning setting [32]. In our context, $\mathcal{X}$ having finite doubling dimension and finite diameter is enough to yield complete coverability ($\varepsilon = 0$ in Definition 4.1).

**Proposition C.4.** *[14, Lemma 4.7] Let $\mathcal{X} \subset \mathbb{R}^d$ have diameter $D$ and doubling dimension $d' = \Theta(d)$. For $T \in \mathbb{N}$, there exists a covering of $\mathcal{X}$ with $N \leq \left(\frac{2D}{r}\right)^{d'}$ balls of radius $\frac{D}{2^T}$.*

**Memorization learning rule.** For completeness we provide pseudocode for the memorization learning rule (Algorithm 4).

**Sample complexity upper bound.** We prove Theorem 4.2.

*Proof of Theorem 4.2.* The idea is that with enough samples every ground-truth positive instance in a ball has a sampled positive instance in the same ball that it can improve to. We first claim that $h$ incurs zero improvement loss on points $x$ for which $f^*(x) = 0$. Since $h(x') = 1 \implies f^*(x') = 1$ by definition of the memorization learning rule, we must have $h(x) = 0$. If there exists $x' \in \Delta(x)$ for which $h(x') = 1$, then $x$ will improve to some such $x'$ which has $h(x') = f^*(x') = 1$. If there is no such $x'$, then $x$ stays put and $h(x) = f^*(x) = 0$. Hence we only need to consider points $x$ for which $f^*(x) = 1$. If $h(x) = 1$ as well then $x$ does not incur error. For all other points $x$ for which $h(x) = 0$ and $f^*(x) = 1$, namely the false negatives, due to the coverability assumption we claim that for all but $\varepsilon$ fraction of these points there will exist a positively labeled sample in every ball that $x$ can improve to. Let balls $B_1, \ldots, B_N$ each with radius $\frac{r}{2}$ and mass $\mathbb{P}_{\mathcal{D}_{\mathcal{X}}^+}[B_k] \geq \beta$ cover $\mathcal{D}_{\mathcal{X}}^+$. Then by Lemma C.2, after $m = O\left(\frac{1}{\beta} \log \frac{N}{\gamma}\right)$ samples, with probability at least $1 - \gamma$ there will be at least one positively labeled sample in every ball $B_k$. This implies that least $1 - \varepsilon$ fraction of points $x \in \mathcal{D}_{\mathcal{X}}^+$ are within distance $r$ of a sampled positive instance $x' \in \Delta(x), f^*(x') = 1$. All such $x$ can improve to $x'$ and hence $\text{Loss}(h, f^*) \leq \varepsilon$ as desired. $\qquad\square$

**Sample complexity lower bound.** We prove Theorem 4.3.

*Proof of Theorem 4.3.* Consider a data distribution $\mathcal{D}$ that consists of $N = \frac{1}{\beta}$ distinct points each with probability mass $\beta$ such that the pairwise distance between points is greater than the improvement radius $r$. Let the concept class $\mathcal{H}$ consist of all $2^N$ possible labelings of these $N$ points, with the rest of $\mathcal{H}$ being labeled negative. By construction $\mathcal{D}$ is trivially $\left(0, \beta, \frac{1}{\beta}\right)$-coverable. We claim that sampling every point is necessary to achieve $\frac{1}{\beta}$ error. Assume for contradiction that there is an unsampled point $x^*$. Since $\mathcal{H}$ consists of all possible labelings of the points, $x^*$ could be labeled negative or positive. Let $h$ be the predictor that the learning algorithm $\mathcal{A}$ outputs and let $f^*$ be the ground-truth concept. We consider two cases. If the predictor $h$ does not label any point $x$ with $\|x - x^*\|_2 \leq r$ positive, then $x^*$ is misclassified when $f^*(x^*) = 1$ since there is no point in $\Delta(x^*)$ that $x^*$ can improve to. Otherwise, the predictor $h$ labels some point $x$ with $\|x - x^*\|_2 \leq r$ positive.

Then when $f^*(x^*) = 0$, $x^*$ will move to $x$, but the ground truth is $f^*(x) = 0$ and hence $x^*$ will be misclassified. Both cases yield a contradiction if there is an unsampled point, thus proving the claim. By a standard coupon-collector analysis, $m = \Omega(N \log N) = \frac{1}{\beta} \log \frac{1}{\beta}$ samples are necessary to sample every point. $\qquad\square$

## D   Learning linear separators optimally under bounded noise

**General reduction to PAC-learning with noise**   We prove Theorem 5.1.

*Proof of Theorem 5.1.* Suppose we are given a noisy sample $S$ with bounded noise $\leq \nu$. To construct the classifier in the improvements setting, we start with a classifier $\hat{h} \in \mathcal{H}$ that achieves low excess error $\varepsilon$ in the standard PAC-learning setting. Using a well-known property of bounded noise (see e.g. Section 5.1 in Balcan and Haghtalab [10]), the disagreement of $\hat{h}$ with the Bayes optimal classifier $f^*_{\text{bayes}}$ can be upper bounded as

$$\Pr_{\mathcal{D}}[\hat{h}(x) \neq f^*_{\text{bayes}}(x)] \leq \frac{\varepsilon}{1 - 2\nu}.$$

Now, for $\mathcal{D}$ isotropic and log-concave, this further implies [43, 10] that there is an absolute constant $C$ such that

$$\theta(\hat{h}, f^*_{\text{bayes}}) \leq C \cdot \Pr_{\mathcal{D}}[\hat{h}(x) \neq f^*_{\text{bayes}}(x)] \leq \frac{C\varepsilon}{1 - 2\nu},$$

where $\theta(h_1, h_2)$ is angle between the normal vectors of the linear separators $h_1$ and $h_2$. We set $\hat{\theta} = \frac{C\varepsilon}{1-2\nu} = r$ and define $\hat{\mathcal{H}} := \{h \in \mathcal{H} \mid \theta(\hat{h}, h) \leq \hat{\theta}\}$. Note that $f^*_{\text{bayes}}(x) \in \hat{\mathcal{H}}$. Now, we define $P := \{x \in \mathcal{X} \mid h(x) = 1 \text{ for each } h \in \hat{\mathcal{H}}\}$, the positive agreement region of classifiers in $\hat{\mathcal{H}}$. We set our improper classifier $\hat{f} = \mathbf{1}[x \in P]$. We will now bound the error $\text{Loss}_{\mathcal{D}, \mathcal{N}}(\hat{f})$ of the above classifier in the improvements setting. Note that if $\hat{f}(x) = 1$ then $f^*_{\text{bayes}}(x) = 1$ by the above construction. These points do not move and the error equals $\Pr_{\mathcal{N}}[f^*_{\text{bayes}}(x) \neq y]$.

If $\hat{f}(x) = 0$, we have two cases. Either, there is a point $x'$ with $\arccos(\langle x, x' \rangle) \leq r$ and $\hat{f}(x') = 1$. In this case, the agent moves to some such $x'$. But since $\hat{f}(x') = 1$, we also have $f^*_{\text{bayes}}(x') = 1$ and again $\Pr_{y'|x' \sim \mathcal{N}}[\hat{f}(x') \neq y'] = \Pr_{y'|x' \sim \mathcal{N}}[f^*_{\text{bayes}}(x') \neq y']$. Else, the distance of $x$ to any positive point $x'$ satisfies $\arccos(\langle x, x' \rangle) > r = \hat{\theta}$. Then, any $h \in \hat{\mathcal{H}}$ (including $f^*_{\text{bayes}}$) must classify $x$ as negative. In this case, the agent does not move and again the error of $\hat{f}$ matches that of $f^*_{\text{bayes}}$.

Put together, the above cases imply that for any point $x$, $\text{Loss}_{\mathcal{N}}(x; h) = \Pr_{\mathcal{N}}[f^*_{\text{bayes}}(x) \neq y]$. $\qquad\square$

## E   Online learning on a graph

**Standard majority vote algorithm fails to learn with improvements.**   We construct an example where the standard majority vote algorithm can result in an unbounded number of mistakes in the learning with improvements setting.

**Example E.1.** *Let $G$ be the star graph, with leaf nodes $x_1, \ldots, x_\Delta$ for $\Delta > 2$, and the center node $x_{\Delta+1}$. Let $\mathcal{H} = \{h_1, \ldots, h_\Delta\}$, where*

$$h_i(x_j) = \begin{cases} \mathbf{1}[i \neq j], & \text{if } j \in [\Delta], \\ 0, & \text{otherwise } (j = \Delta + 1). \end{cases}$$

*Let $f^* = h_1$. The standard majority vote algorithm uses a majority vote to make the prediction at each node. At time $t = 1$, say the learner sees the center node $x^{(1)} = x_{\Delta+1}$. We have that*

$$\hat{h}^{(1)}(x_j) = \begin{cases} 1, & \text{if } j \in [\Delta], \\ 0, & \text{otherwise } (j = \Delta + 1). \end{cases}$$

*The learner suffers a mistake, as there is $x' = x_1$ such that $f^*(x') = 0$ but $\hat{h}^{(1)}(x') = 1$. However, all the classifiers agree with $f^*$ on $x_{\Delta+1}$, and no classifier is discarded. Thus, if the online sequence of nodes simply consists of repeated occurrences of the center node, that is, $x^{(t)} = x_{\Delta+1}$ for all $t$, then the learner using the standard majority vote algorithm suffers a mistake on every round.*

**Mistake upper bound in realizable setting.** We prove Theorem 6.1.

*Proof of Theorem 6.1.* Suppose there is a mistake on round $t$. If $\hat{h}^{(t)}(x^{(t)}) = f^*(x^{(t)}) = 1$, there is no mistake as the agent does not move at all. We consider three cases.

- **Case 1:** If $\hat{h}^{(t)}(x^{(t)}) = 1, f^*(x^{(t)}) = 0$, then the agent doesn't move, and we discard classifiers $h \in H$ that predict $h(x^{(t)}) = 1$. Clearly, $f^*$ is not discarded, and we discard at least $\frac{\Delta_G}{\Delta_G+1}|H|$ classifiers that make a mistake on $x^{(t)}$.
- **Case 2:** If $\hat{h}^{(t)}(x^{(t)}) = 0, f^*(x^{(t)}) = 0$, then there must be some $x' \in \Delta(x^{(t)})$ such that $\hat{h}^{(t)}(x') = 1, f^*(x') = 0$. This implies that for each such $x'$, we must have $|\{h \in H \mid h(x') = 0\}| < \frac{|H|}{\Delta_G+1}$. Taking a union over all neighbors in $\Delta^+$, we conclude that $|H'| \geq \frac{|H|}{\Delta_G+1}$.
- **Case 3:** If $\hat{h}^{(t)}(x^{(t)}) = 0, f^*(x^{(t)}) = 1$, then the agent can potentially move. If it does not move, then it must be the case that $|\Delta^+| = 0$, else $x^{(t)}$ would have moved and changed the predicted label to 1. In this case, we discard at least $\frac{|H|}{\Delta_G+1}$ classifiers $h \in H$ that predict $h(x^{(t)}) = 0$. It is however also possible that the agent moves and still makes a mistake. In this case, $|\Delta^+| > 0$ but there is some point $x' \in \Delta(x^{(t)})$ in the neighborhood of $x^{(t)}$ such that $\hat{h}^{(t)}(x^{(t)}) = 1$ but $f^*(x^{(t)}) = 0$. In this case our algorithm discards $H'$, the set of classifiers that predict positive on all neighbors of $x^{(t)}$, and as argued above, $|H'| \geq \frac{|H|}{\Delta_G+1}$.

Thus we discard at least a $\frac{|H|}{\Delta_G+1}$ fraction of the classifiers on each mistake, implying the desired mistake bound. Indeed, since $f^*$ never gets discarded and we are in the realizable setting, if we make $M$ mistakes then $\left(1 - \frac{1}{\Delta_G+1}\right)^M |\mathcal{H}| \geq 1$, or $M \leq -\frac{\log|\mathcal{H}|}{\log\left(1-\frac{1}{\Delta_G+1}\right)} \leq (\Delta_G + 1)\log|\mathcal{H}|$. $\square$

**Mistake upper bound in agnostic setting.** We prove Theorem 6.2.

*Proof of Theorem 6.2.* The overall arguments are similar to those in the proof of Theorem 6.1. Suppose there is a mistake on round $t$. We have the following three cases.

$\hat{h}^{(t)}(x^{(t)}) = 1, f^*(x^{(t)}) = 0$. In this case, we halve the weights $w_h$ for classifiers $h \in H$ that predict $h(x^{(t)}) = 1$. The reduction in the total weight $W_t$ in this case is at least $\frac{W_t^+}{2} \geq \frac{\Delta_G}{2(\Delta_G+1)}W_t$. That is, $W_{t+1} \leq W_t - \frac{\Delta_G}{2(\Delta_G+1)}W_t \leq W_t\left(1 - \frac{1}{2(\Delta_G+1)}\right)$.

$\hat{h}^{(t)}(x^{(t)}) = 0, f^*(x^{(t)}) = 0$. Since the learner made a mistake, there must be some $x' \in \Delta(x^{(t)})$ such that $\hat{h}^{(t)}(x') = 1, f^*(x') = 0$. For each such $x'$, the learner predicted positive and so $\sum_{h \in \mathcal{H}|h(x')=1} w_h \geq \frac{\Delta_G}{\Delta_G+1}W_t$, or $\sum_{h \in \mathcal{H}|h(x')=0} w_h \leq \frac{W_t}{\Delta_G+1}$. Taking a union over all neighbors in $\Delta^+$, we get $\sum_{h \in H'} w_h \geq W_t - |\Delta^+|\frac{W_t}{\Delta_G+1} \geq W_t - \Delta_G\frac{W_t}{\Delta_G+1} = \frac{W_t}{\Delta_G+1}$. By Line 10 of Algorithm 3, this implies that $W_{t+1} \leq W_t\left(1 - \frac{1}{2(\Delta_G+1)}\right)$.

$\hat{h}^{(t)}(x^{(t)}) = 0, f^*(x^{(t)}) = 1$. If the agent does not move, then $|\Delta^+| = 0$. In this case, we halve the weight of all classifiers that predicted negative on $x^{(t)}$. Since $\hat{h}^{(t)}(x^{(t)}) = 0$ implies that $W_t^+ < \frac{\Delta_G}{\Delta_G+1}W_t$, $W_{t+1} < W_t\left(1 - \frac{1}{2(\Delta_G+1)}\right)$. On the other hand, if the agent moves, then $|\Delta^+| > 0$. We discard $H'$ and as shown in the previous case, $W_{t+1} \leq W_t\left(1 - \frac{1}{2(\Delta_G+1)}\right)$.

Thus, in all cases when there is a mistake, $W_{t+1} \leq W_t\left(1 - \frac{1}{2(\Delta_G+1)}\right)$. Thus, after $M$ mistakes, $W_t \leq |\mathcal{H}|\left(1 - \frac{1}{2(\Delta_G+1)}\right)^M$. Since $f^*$ makes at most OPT mistakes, we have $W_t \geq \frac{1}{2^{\text{OPT}}}$. Putting together, $\frac{1}{2^{\text{OPT}}} \leq |\mathcal{H}|\left(1 - \frac{1}{2(\Delta_G+1)}\right)^M$, which simplifies to the desired mistake bound. $\square$

**Mistake lower bounds in realizable and agnostic settings.** We prove Theorem 6.3.

*Proof of Theorem 6.3.* The proof is analogous to [3, Theorem 4.7] which shows lower bounds under strategic online classification. For $\Delta > 1$ we use a star graph $G$ with leaf nodes $x_1, \ldots, x_\Delta$ for $\Delta > 2$ and the center node $x_{\Delta+1}$. Let $\mathcal{H} = \{h_1, \ldots, h_\Delta\}$, where $h_i(x_i) = 1$ and $h_i(x_j) = 1$ for all other $j \neq i$.

We first consider the agnostic setting. The proof idea is to construct an adaptive adversary that chooses an example that induces a mistake at *every* round, but such that at every round all but at most one hypothesis from $\mathcal{H}$ can classify this example correctly. Formally, after the learning algorithm outputs the predictor $h^{(t)}$ at round $t$ the adversary chooses the next example according to the following procedure:

- **Case 1:** If $h^{(t)}(x_{\Delta+1}) = 1$, then the adversary picks the labeled example $(x_{\Delta+1}, 0)$ and ground-truth labeling

$$f^*(x_j) = \begin{cases} 1 & \text{if } j \in [\Delta] \\ 0 & \text{if } j = \Delta + 1. \end{cases}$$

  Since $x_{\Delta+1}$ does not move under $h^{(t)}$ then $h^{(t)}$ fails to classify $x_{\Delta+1}$ correctly. On the other hand, every classifier $h_i \in \mathcal{H}$ classifies $x_{\Delta+1}$ correctly since under $h_i$, $x_{\Delta+1}$ will move to the point $x_i$ which has positive label under both $h_i$ and $f^*$.

- **Case 2:** If $h^{(t)}(x_j) = 0$ for all $j \in [\Delta+1]$, then the adversary picks the labeled example $(x_{\Delta+1}, 1)$ and the ground-truth labeling $f^*(x_j) = 1$ for all $j \in [\Delta + 1]$. Since all points have negative label under $h^{(t)}$, $x_{\Delta+1}$ cannot improve under $h^{(t)}$, so $h^{(t)}$ fails to classify $x_{\Delta+1}$ correctly. On the other hand, every classifier $h_i \in \mathcal{H}$ classifies $x_{\Delta+1}$ correctly since under $h_i$, $x_{\Delta+1}$ will move to the point $x_i$ which has positive label under both $h_i$ and $f^*$.

- **Case 3:** If $h^{(t)}(x_{\Delta+1}) = 0$ and $h^{(t)}(x_j) = 1$ for some $i \in [\Delta]$, then the adversary picks the labeled example $(x_j, 0)$. Since $x_i$ does not move under $h^{(t)}$ then $h^{(t)}$ fails to classify $x_j$ correctly. On the other hand, every classifier $h_i \in \mathcal{H}$ for $i \neq j$ classifies $x_j$ correctly as negative since $x_j$ cannot improve.

By the above analysis, $h^{(t)}$ makes a mistake on the next example for all $t$. However, for each $t$ at most one hypothesis from $\mathcal{H}$ makes a mistake, implying that the sum of the number of the mistakes made by all hypotheses over all rounds is at most the current round number $t$. Since $|\mathcal{H}| = \Delta$ by the pigeonhole principle there exists a hypothesis that makes at most $\frac{t}{\Delta}$ mistakes, so $\mathsf{OPT} \leq \frac{t}{\Delta}$, implying that the number of mistakes made is $t \geq \Delta \cdot \mathsf{OPT}$ in the agnostic setting.

For the proof of the $\Delta - 1$ lower bound in the realizable setting, we use the same construction as in the agnostic setting but restrict to $t = \Delta - 1$ rounds. The learning algorithm makes $\Delta - 1$ mistakes, but at this point there still exists at least one hypothesis that has made no mistakes so far, say $h_i$ for $i \in [\Delta]$. Then the adversary can keep using $h_i$ as the ground truth for the first $\Delta - 1$ rounds in the above procedure so that the ground truth is still realizable after $\Delta - 1$ rounds. $\qquad \square$

