# OpenReview forum: "Conservative classifiers do consistently well with improving agents: characterizing statistical and online learning"
_NeurIPS.cc/2025/Conference — NeurIPS 2025 spotlight_

### Official Review · Reviewer_78PX · 2025-06-19

**Clarity:** 3
**Significance:** 3
**Originality:** 3
**Rating:** 5
**Confidence:** 4

**Summary:**

This paper continues the study of the setting known as "learning with improvements." Here, agents (i.e. instances of the example space) can react to a published classifier by moving to a new instance which is positively classified by the published classifier. Compared to "strategic classification", in the "learning with improvements" setting,  agents can genuinely improve in order to attain the desirable classification. The authors provide several contributions. They provide an exact characterization of proper learning with improvements in the realizable setting in terms of a variant of minimally consistent concept classes. In the improper setting, the authors characterize learnability with improvements under a mild generative assumption on the data distribution. Finally, the authors also provide results under label noise as well as in the online setting.

**Questions:**

Suggestions:
1) I think it would be good to provide some more intuition about when the loss in Equation (2.1) is 1, especially in terms of false positives and false negatives. In particular, it seems like the loss is 1 as long as there exists a false positive in $\Delta_h(x)$ or h(x) is a false negative.
2) In lines 140-146, I think it would be helpful to define "least consistent" explicitly in terms of the support of a hypothesis $h \in H$ defined as $\supp(h) = \{x: h(x) = 1\}.$ That is, for a realizable sample $S$, a hypothesis $g \in H$ is least consistent, if for every consistent hypothesis $h \in H$, we have that $\supp(g) \subseteq \supp(h)$.
3) In lines 140-146 and Definition 3.3, I think it would be good to give some examples of classes that are minimally and nearly minimally consistent.
4) $\Delta_G$ in Theorem 6.1 needs to be defined (I think it's the max out degree)

Questions:
1) Theorem 3.4 seems to be a qualitative characterization of proper learning with improvements. Is there a quantitative characterization? That is, what is the optimal sample complexity for learning with improvements? Is there a dimension that characterizes this?
2) Perhaps I'm mistaken, but when learning with label-noise, the goal is still to obtain small population error with respect to the true labels (not the noisy labels). It's just that the labels in the training data are noisy. This makes me confused about the noisy loss function defined in Line 274. Why is the loss here in terms of the noisy labels? What does getting a loss of $1$ here mean?
3) In Lines 193-194, are you claiming that proper learning is without loss of generality if $\Delta$ can be anything? Are you claiming that there exists a choice of $\Delta$ such that a class $H$ is improperly learnable with improvements if and only if it is nearly minimally consistent?
4) The algorithms for online learning are deterministic. While this is fine in the realizable setting, in the agnostic setting, randomization is key to removing the multiplicative factor in front of OPT. Is there a natural way to randomize Algorithm 3 and remove the $\Delta$ factor in front of OPT? If not, what is the barrier?

**Ethical Concerns:**

["NO or VERY MINOR ethics concerns only"]

**Final Justification:**

I maintain my positive score as the authors have adequately answered my questions.

**Limitations:**

yes

**Paper Formatting Concerns:**

no concerns

**Quality:**

3

**Strengths And Weaknesses:**

Strengths:
- This paper is well written and easy to follow
- The problem setting is clean and practically relevant
- In my opinion, the results are interesting and present a clear contribution to the learning theory community.

Weaknesses:  No major weaknesses stand out to me. That said, I would appreciate it if the authors could address my questions/suggestions below, as there are a few places where I believe that readability can be improved.

---

> ### Author Rebuttal · Authors · 2025-07-31
>
> We thank the reviewer for their comments and helpful suggestions (which we intend to incorporate in our revision), and address their questions and concerns below.
>
> 1. Theorem 3.4 seems to be a qualitative characterization of proper learning with improvements. Is there a quantitative characterization? That is, what is the optimal sample complexity for learning with improvements? Is there a dimension that characterizes this?
>
> Yes, the proof of Theorem 3.4 in Appendix B gives a quantitative characterization. The sample complexity is governed by the (vanilla) VC-dimension when the concept class is learnable at all with improvements. Specifically, when $\mathcal{H}$ is nearly minimally consistent, the sample complexity is that same as that of vanilla PAC-learning: $O(\frac{1}{\varepsilon} (d_\text{VC}(\mathcal{H}) + \log \frac{1}{\delta}))$  under the realizability assumption (see bottom of page 14), and this is tight because the same lower bond argument for vanilla PAC-learning also serves as a lower bound for learning with improvements (take the closure of the concept class to make it minimally consistent). On the other hand, when the concept class is not minimally consistent, the proof of Theorem 3.4 constructs an improvement function $\Delta$ (recall that the definition of learnability in Section 3 is respect to every improvement function) and a distribution $\mathcal{D}$ for which any learner (even computationally unbounded) suffers constant error, so the sample complexity is unbounded. (In particular, a concept class could have small VC-dimension and thus small vanilla sample complexity but if it is not nearly minimally consistent it is guaranteed to have unbounded sample complexity in the improvements model.) To summarize, while VC-dimension is not sufficient to guarantee learnability with improvements, when the concept class is nearly minimally consistent, the sample complexity is governed by the VC-dimension.
>
> 2. Perhaps I'm mistaken, but when learning with label-noise, the goal is still to obtain small population error with respect to the true labels (not the noisy labels). It's just that the labels in the training data are noisy. This makes me confused about the noisy loss function defined in Line 274. Why is the loss here in terms of the noisy labels? What does getting a loss of $1$ here mean?
>
> Great question. While it seems sensible to define the loss in terms of the true labels, the standard literature on bounded noise (e.g. Balcan and Haghtalab [2020], Diakonikolas et al. [2019]) define error as the expected loss of the classifier over the joint (noisy) distribution $\mathcal{D}$ over $X\times Y$, and the goal is to compete with the Bayes optimal classifier which predicts the most likely label for each point (the bounded noise is assumed to be less than $\frac12$ so this matches the “true” label without noise) and whose expected error is given by "OPT" (which is no more than the noise upper bound $\nu$). The guarantees in standard PAC learning with noise (i.e. with improving or strategic agents) are usually stated in terms of excess expected error over “OPT" in terms of a loss function that includes an expectation over the noise. Specifically in our setting, in addition to following the standard PAC learning convention, we choose the expected loss as it has the following nice interpretation. In the noisy setting, whether an agent is truly qualified given features $x'$ is a random variable (as opposed to a deterministic function $f^* (x)$ in the noiseless setting), and we want our predictions to be accurate (even as the agents move) in expectation. A loss of $1$ means that there is some point $x’$ to which the agent can “improve” to (if negative, the agent stays put if $h(x)=1$) such that our prediction $h(x)$ is completely inaccurate (almost surely).
>
> 3. In Lines 193-194, are you claiming that proper learning is without loss of generality if $\Delta$ can be anything? Are you claiming that there exists a choice of $\Delta$ such that a class $H$ is improperly learnable with improvements if and only if it is nearly minimally consistent?
>
> Here is a clearer wording of that paragraph that we just edited based on your feedback:
>
> We first note that in order to get interesting results, we must make some assumption on the improvement function $\Delta$. This is because we show that any concept class that is improperly PAC-learnable for all $\Delta$ must also be properly PAC-learnable, and we have already characterized proper PAC-learnability in Section 3 using the nearly minimally consistent property. (Recall that proper learnability trivially implies improper learnability.) Indeed, if $\Delta(x) = \mathcal{X}$ for all $x\in \mathcal{X}$ then the learning algorithm cannot make any false positives or else every negative instance can "improve" to such a false positive and incur a classification error. On the other hand, if $\Delta(x) = \\{x\\}$ for all $x\in \mathcal{X}$ then the learning algorithm cannot incur too many false negatives as any false negative is unable to improve. These two examples show that if we make no assumption on $\Delta$, the concept class must be learnable with one-sided error, which implies that it is properly PAC-learnable by our results in Section 3.
>
> 4. The algorithms for online learning are deterministic. While this is fine in the realizable setting, in the agnostic setting, randomization is key to removing the multiplicative factor in front of OPT. Is there a natural way to randomize Algorithm 3 and remove the $\Delta$ factor in front of OPT? If not, what is the barrier?
>
> We believe that it is indeed possible to give a randomized algorithm which achieves sub-linear regret (as opposed to $\Delta_G \cdot \text{OPT}$ mistakes). One way to do this is to use a modified version of the Hedge algorithm (or EXP3 under bandit feedback) where we update the weights after $K$ rounds and use the all-negative classifier on one of every $K$ rounds (chosen uniformly randomly). This would be a conservative version of Algorithm 3 of Ahmadi et al. (2023), a difference being they use the all-positive classifier instead to incentivize the agent to not make a deceptive move (while we use a conservative all-negative classifier to keep the agent from trying to improve).

---

> ### Comment · Reviewer_78PX · 2025-08-04
>
> I thank the authors for their response. Looking at the proof of Theorem 3.4 and the authors reply to my first question, it seems like you require the class H to also have finite VC dimension. However, the fact that H needs to have finite VC dimension is not stated in Theorem 3.4. Can the author's clarify this? Should the correct characterization of proper PAC-learnable with improvements be:
>
> "A class H is properly PAC-learnable with improvements if and only if H has finite VC dimension and is nearly minimally consistent"?
>
> If so, your proof of the necessity of finite VC dimension in your reply to Question 1 should be included in the paper and the statement of Theorem 3.4 needs to be updated accordingly.

---

> > ### Author Response · Authors · 2025-08-05
> >
> > Yes, that's right, thanks a lot for bringing this to our attention! The concept class does need to have finite VC-dimension, and we will change the statement to:
> >
> > A concept class $\mathcal{H}$ is properly PAC-learnable with improvements for all improvement functions and all data distributions if and only if $\mathcal{H}$ has finite VC-dimension and is nearly minimally consistent.
> >
> > And we will also add the straightforward argument for the necessity of finite VC-dimension, namely that PAC learning with improvement reduces to standard PAC learning when $\Delta(x)=\\{x\\}$ for all points $x$.
> >
> > In addition we also added the finiteness of VC-dimension as an assumption to the theorem from Natarajan's textbook
> > that minimally consistent characterizes learning with one-sided error (the textbook also omitted this assumption in the statement of their theorem, and it was implicitly assumed in the Natarajan text and our statement):
> >
> > A concept class is PAC-learnable with one-sided error if and only if it has finite VC-dimension and is minimally consistent.

---

> > > ### Comment · Reviewer_78PX · 2025-08-05
> > >
> > > I thank the authors for their response and will maintain my positive score.

---

### Official Review · Reviewer_Zw74 · 2025-07-02

**Clarity:** 4
**Significance:** 3
**Originality:** 3
**Rating:** 5
**Confidence:** 3

**Summary:**

* This paper extends the “learning with Improvements” framework (Attias et al., ICML 2025) to more general settings.
* In the learning with improvements setting, feature-label pairs represent agents, which respond to deployed classifiers. Given classifier $h$, agents represented by a feature vector $x$ move to a point in their “improvement set” $x’\\in\\Delta(x)$ if their classification can be made positive ($h(x)=0$, $h(x’)=1$), and stay in place otherwise. True label $f^*(x’)$ is a function of the “improved” feature vector $x’$, and the classifier’s performance is measured according to its post-response accuracy, with worst-case tie breaking. A learning algorithm has access to a finite sample from a base distribution, and the goal is to prove statistical learnability.
* The main theoretical result of Section 3 is a characterization: A class $\\mathcal{H}$ is PAC-learnable with improvements if and only if it is nearly minimally consistent - a variation of the known notion of minimal consistency, which characterizes PAC learnability with one-sided error.
* Section 4 explores improper learnability when the improvement functions are the set of euclidean balls. The main positive result gives a learnability guarantee that depends on distribution coverability, and the main negative result provides a lower bound for the required training set size in the general case.
* Section 5 explores the learnability of linear separators under bounded label noise, relating sample complexity of upper bounds for learning with improvements in this setting to bounds of PAC learnability in the bounded noise model.
* Finally, Section 6 explores learning with improvements in the online discrete-graph setting, providing positive results for realizable and agnostic settings.

**Questions:**

* Is it possible to extend the characterization result in Section 3 to the non-realizable case?
* How would the results change if the agent is not assumed to make the "worst case" choice among all points in $\\Delta(x)$?
* Are there classes $\\mathcal{H}$ which do allow for proper learning under ball improvement sets?
* What are the possible societal implications of the results, or in which way results can be extended to increase societal impact?

**Ethical Concerns:**

["NO or VERY MINOR ethics concerns only"]

**Final Justification:**

The paper is very interesting, and the clarifications during the rebuttal were very helpful. I maintain my original score, and I believe that the paper can be further strengthened by expanding the discussion of possible societal implications.

**Limitations:**

Relation to adjacent work is discussed, and questions that arise from the analysis are stated clearly.

**Paper Formatting Concerns:**

No concerns.

**Quality:**

3

**Strengths And Weaknesses:**

Strengths:
* Significantly extends a set of recently published results in a meaningful way.
* Presentation is clear and easy to follow. Helpful intuition for proofs is provided.
* Relation to adjacent work is clearly explained, both formally and intuitively.

Weaknesses
* No empirical evaluation.
* Possible societal impact is not discussed.

---

> ### Author Rebuttal · Authors · 2025-07-31
>
> We thank the reviewer for their comments and address their questions and concerns below.
>
> 1. Is it possible to extend the characterization result in Section 3 to the non-realizable case?
>
> As far as we know it is not possible to extend the result in Section 3 to the non-realizable setting. In general it seems hard to get results for agnostic learning in this model, but we are able to get a nontrivial agnostic learning sample complexity upper bound for the discrete graph model in Section 6.
>
> 2. How would the results change if the agent is not assumed to make the "worst case" choice among all points in $\Delta(x)$?
>
> This is a promising direction for future work. Indeed we assume that the agent can make the “worst case” choice in order to make our results robust to the agent’s choice (since they don’t know which points are ground truth positive or negative). We did consider exploring what happens if the agent for example moves to a “random” positive point according to $h$, but this becomes hard to analyze mathematically especially if the improvement set is a continuous region and not discrete points.
>
> 3. Are there classes $\mathcal{H}$ which do allow for proper learning under ball improvement sets?
>
> Theorem 3.4 shows that any nearly minimally consistent concept class is properly learnable for all improvement sets and in particular Euclidean balls. This is actually a fairly broad set of concept classes, for example it includes (but is not limited to) all intersection-closed classes such as intervals on a line, rectangles in $\mathbb{R}^2$, polytopes in $\mathbb{R}^d$, etc.
>
> 4. What are the possible societal implications of the results, or in which way results can be extended to increase societal impact?
>
> One possible societal lesson from the results is that conservative classifiers perform consistently well when agents can improve. In particular, this means that when setting cutoffs, say for college entrance exams, it is often not harmful to increase the cutoff by a bit because qualified students will always be able to work a bit harder to meet it. On the other hand, it can be very harmful to have the cutoff even slightly below the true cutoff (something that is not usually a problem in standard PAC learning). Quantitatively the paper shows that you can even achieve zero classification error in the learning with improvements model.

---

> > ### Comment · Reviewer_Zw74 · 2025-08-06
> >
> > Thank you for the response! I have no further questions.

---

### Official Review · Reviewer_dh1P · 2025-07-03

**Clarity:** 3
**Significance:** 3
**Originality:** 3
**Rating:** 5
**Confidence:** 2

**Summary:**

This paper studies PAC-learnability in the learning with improvement setting, where the agent can improve their feature vectors after observing the hypotheses given by the learner. The authors provide new conditions for which concept class, data distribution is learnable under different conditions, and provide the corresponding learning algorithms. See the "Strengths And Weaknesses" for more discussion on the results.

**Questions:**

The questions are given in "Strengths And Weaknesses".

**Ethical Concerns:**

["NO or VERY MINOR ethics concerns only"]

**Limitations:**

Please see "Strengths And Weaknesses" for more discussion.

**Quality:**

3

**Strengths And Weaknesses:**

Strengths And Weaknesses
---

Significance/novelty of the results:

This paper provides new and significant results for understanding PAC-learnable with impartments, including conditions for learnable concepts, learnable distributions, and more general settings. More specifically:

Without any assumption on the improvement function of the agents, the authors show that the necessary and sufficient condition for PAC-learnability with improvement is the concept class being nearly minimally consistent.

Assuming the agents can only improve their decisions over a ball, the authors further show that there exists an algorithm that can learn any agent distribution if the conditional distribution of positive instances is recoverable.

The authors also consider the more general setting with label noise, and e construct optimal algorithms for learning linear separators. Finally, they also consider online learning with improvements for the discrete graph model, and provide algorithms with near-optimal mistake bounds.


Presentation:

This paper is clearly written and though not very easy to follow. It would be great to shortly discuss/define things like minimally consistent in the introduction.


Questions and comments:


For the setting in ball improvement set discussed in section 4, if X is bounded, how is the improvement set defined for the points at the surface? Is it the interaction between X and Delta(x)?

In Definition 4.1,  D is (epsilon, beta, N) coverable if … radius r. why (epsilon, beta, N) coverable does not depend on r? (e.g., (epsilon, beta, N, r) coverable). Also, why do Theorems 4.2 and 4.3 not depend on r? What if r is infty then?


Theorem 3.4 shows a necessary and sufficient condition for PAC-learnable with improvements. However, it seems to be transforming from one definition to another. While it is nice, I wonder if the authors could discuss what it implies? How does nearly minimally consistent help understanding PAC-learnable with improvements?

Personally, I think this paper would fit COLT or ALT more, but NeurIPS is also suitable.

---

> ### Author Rebuttal · Authors · 2025-07-31
>
> We thank the reviewer for their comments and address their questions and concerns below.
>
> 1. For the setting in ball improvement set discussed in section 4, if X is bounded, how is the improvement set defined for the points at the surface? Is it the interaction between X and Delta(x)?
>
> Yes, it would be the intersection between $\mathcal{X}$ and the ball of radius $r$ centered at $x$. We meant to write $\Delta(x) = \\{ x \in \mathcal{X}, \ldots \\} $, thanks for catching the typo!
>
> 2. In Definition 4.1, D is (epsilon, beta, N) coverable if … radius r. why (epsilon, beta, N) coverable does not depend on r? (e.g., (epsilon, beta, N, r) coverable). Also, why do Theorems 4.2 and 4.3 not depend on r? What if r is infty then?
>
> The parameter $\beta$, which is the probability mass of each ball, implicitly depends on $r$. For example in real applications, if for example the data distribution is uniform over a bounded set in $\mathbb{R}^d$, then a ball of radius $r$ has probability mass proportional to its volume, which is proportional to $r^d$. A larger $r$ (which you can think of as larger $\beta$) actually makes the sample complexity better (the bounds in Theorem 4.2 and 4.3 scale with $1/\beta$), and this can intuitively be seen as well since you can cover the space with less balls and you only need to sample a point in each ball. In the extreme case of $r = \infty$, as soon as there is a positive example $x$ in the training set you can achieve zero error by outputting a hypothesis that is 1 on $x$ and 0 everywhere else, and all other points will improve to $x$.
>
> 3. Theorem 3.4 shows a necessary and sufficient condition for PAC-learnable with improvements. However, it seems to be transforming from one definition to another. While it is nice, I wonder if the authors could discuss what it implies? How does nearly minimally consistent help understanding PAC-learnable with improvements?
>
> The implication is that you can check whether a concept class is learnable with improvements by checking a concrete property of the concept class (nearly minimally consistent) unlike the definition of PAC-learnability which deals with all improvement functions and all data distributions. This is the same type of result as the Fundamental Theorem of Statistical Learning, which says that a concept class is PAC-learnable (defined over all data distributions) if and only if it has finite VC-dimension. We show in the proof of Theorem 3.4 that the sample complexity in learning with improvements is also governed by the VC-dimension, but in addition the concept class must be nearly minimally consistent to be learnable.

---

> > ### Comment · Reviewer_dh1P · 2025-08-05
> > **Response**
> >
> > I would like to thank the authors for the reply and I do not have further questions.

---

### Official Review · Reviewer_9Ssd · 2025-07-04

**Clarity:** 3
**Significance:** 2
**Originality:** 3
**Rating:** 4
**Confidence:** 2

**Summary:**

This paper studies the problem of learning in settings where agents can take improvement actions that beneficially alter their features after observing a published classifier. These improvements are assumed to reflect genuine increases in quality, as opposed to adversarial manipulation commonly studied in strategic classification. The paper builds on prior work by Attias et al. (2025) and investigates several learning settings under this assumption, including proper PAC learning, improper learning with geometric constraints, learning under label noise, and online learning on discrete improvement graphs.

Key contributions include a characterization of proper PAC learnability via a condition termed nearly minimal consistency, sample complexity results for improper learning under Euclidean improvement sets with a coverability assumption, algorithms for noise-robust learning achieving Bayes optimality under specific distributional assumptions, and mistake-bound guarantees for online learning in realizable and agnostic settings. The paper also draws connections to other models in the literature, including PAC learning with one-sided error and strategic classification.

**Questions:**

1. Can you formalize the interaction between the learner and agents using a Stackelberg framework, and explicitly define agent best responses?

2. Please clarify how your model compares to strategic classification—can agents both game and improve, or are improvements enforced by assumption?

3. Can you add examples that illustrate how your algorithms behave on simple input spaces (e.g., linearly separable data under improvements).

4. Can you provide experimental results on synthetic or real datasets to validate your theoretical claims?

**Ethical Concerns:**

["NO or VERY MINOR ethics concerns only"]

**Final Justification:**

The reviewers have addressed most of my questions in the rebuttal

**Limitations:**

not beyond the weaknesses and questions

**Quality:**

3

**Strengths And Weaknesses:**

Strengths
1. Addresses a learning setting distinct from adversarial manipulation by considering beneficial improvements.
2. Provides formal characterizations and bounds in multiple learning paradigms.
3. Extends prior work and resolves some theoretical questions.

Weaknesses
1. Lack of Game-Theoretic Modeling Clarity:
The paper does not explicitly present the learner-agent interaction as a Stackelberg game, nor does it characterize the agent’s best-response behavior given a published classifier.
There is no analysis or modeling of how agents decide to take improvement actions, or under what utility assumptions they operate.

2. Limited Connection to Strategic Learning Literature:
Strategic learning frameworks often distinguish between gaming and improvement actions. This distinction is not discussed, nor is there a clear explanation of when or why the improvement model applies instead of a more general strategic model.

3. Illustrative Examples Missing:
There are no concrete examples or visual illustrations of key ideas (e.g., how classifiers behave under improvements, what consistency conditions mean, or how improvements affect learning performance), which makes some theoretical claims hard to interpret.

4. No Empirical Evaluation:
The paper does not include experiments on real or synthetic data. As a result, it is unclear whether the theoretical findings lead to effective or robust learning performance in practice.
There is no evidence provided to support the practical efficiency, robustness, or scalability of the proposed algorithms.

5. It is difficult to tell the difficulty/novelty of the proof techniques

---

> ### Author Rebuttal · Authors · 2025-07-31
>
> We thank the reviewer for their comments and address their questions and concerns below.
>
> 1. Can you formalize the interaction between the learner and agents using a Stackelberg framework, and explicitly define agent best responses?
>
> As stated in the introduction and Section 2, the formal model is the following: first the learner publishes a classifier $h$, and the agents best respond to this classifier by moving to an arbitrary point $x’$ in the improvement set $\Delta(x)$ such that $h(x’) = 1$ since they want to be positively classified. The classification error is computed based on whether or not $x’$, the point that $x$ moved to, is actually positive according to the ground truth $f^* $. We assume adversarial tie breaking, that is if an agent can move to a point which gets them a positive classification $h(x’) = 1$ but without being truly qualified $f(x’)=0$, then it will prefer to do that. Note that this basic formulation itself is not novel: it was introduced by Attias et al. (ICML 2025), and they also establish the game-theoretic connection.
>
> It is straightforward to view this in the Stackelberg framework. Simply set the utility of the agent to be $1$ at positive points $h(x)=1$, $0$ at negative points $h(x)=0$ and the utility of movement to a point $x'$ within  $\Delta(x)$ is something in $[0,1]$. As an example, we could set the utility of movement to $\mathbb{I}[x’ \in \Delta(x)] \cdot h(x’) \cdot \(2 - f^* (x’)\) / 3$. This utility is such that the agent has no incentive to move if either $\mathbb{I}[x’ \notin \Delta(x)]$ or $h(x’)=0$, else it has a utility of $\frac23$ for actually negative points $f^* (x’)=0$ and $\frac13$ for actually positive points (so it moves in either case but prefers the former). Note that $\Delta(x)$ is the set of points to which the agent $x$ would ever consider (or is able to) move to, and its incentives to move to a point $x’$ within $\Delta(x)$ are governed by $h(x’)$ and $f^* (x’)$ as described above.
>
> Conversely, our improvement set based abstraction can be used to model an arbitrary utility function as follows. The utility of being positive is $1$ and negative is $0$ (without loss of generality). Movements are associated with cost functions (negative utilities) for each $x\to x'$ move, and costs outside of $(0,1)$ can be ignored (agent either always moves or never moves, irrespective of the classifier). We can define the set $\Delta(x)$ to be the set of points where the cost is in $(0,1)$. Now the agent with $h(x)=0$ would want to move to a point with $h(x’)=1$ as long as it is within $\Delta(x)$, as the net utility is $1 - \text{cost}(x,x’)$, which is positive. We can further incorporate worst-case tie-breaking by defining $\Delta(x)$ to only consist of points with $f^* (x)=0$. We are happy to add this equivalence as a remark (this is alluded to but not explicitly stated).
>
> 2. Please clarify how your model compares to strategic classification—can agents both game and improve, or are improvements enforced by assumption?
>
> As stated in the introduction, the agent can move to any point in the improvement set. We call this “improving” rather than “gaming” (the latter term is used in strategic classification) because the agent is genuinely changing their features. What do you mean by “improvements enforced by assumption”? It is not enforced that agents move, but naturally they want to be positively classified so they will of course move to a positive point. Finally, we respectfully disagree with the comment that there is limited comparison to the strategic learning literature and note that we have indeed compared all the results in our model to strategic classification. Every section of our results has a subsection/paragraph comparing the results to strategic classification and we have contextualized the differences in our results on learning with improvements with respect to learnability and algorithm design in strategic classification. See lines [178-186], [249-257], [303-310], [337-340] for example.
>
> 3. Can you add examples that illustrate how your algorithms behave on simple input spaces (e.g., linearly separable data under improvements).
>
> Yes, we will add these examples to the paper! It must also be specified what the improvement set is. As an example, in the linearly separable case with Euclidean ball improvement sets, our results in Section 3 and 4 demonstrate that a good classifier should be the smallest halfspace that contains all the positive training examples. If the improvement radius is at least the margin between negatively and positively labeled examples, then one can even get zero error by using this classifier.
>
> 4. Can you provide experimental results on synthetic or real datasets to validate your theoretical claims?
>
> This is a theoretical machine learning paper. The results in the paper establish upper and lower bounds on sample complexity and mistake bounds of learning with improvements, which we believe are not suitable to verify experimentally but rather through proofs. Our techniques may inspire future empirical research, for example more conservatively designed classifiers with respect to false positives, to achieve better empirical effectiveness in the presence of improving agents.

---

> > ### Comment · Area_Chair_k4SS · 2025-08-05
> >
> > Dear reviewer 9sSd,
> >
> > The authors have provided detailed responses to the points you mentioned. Please review the author's response to check if the concerns you mentioned are addressed, and follow up as necessary so that authors have a chance to respond. Please do not wait till the last minute, and also do not forget to acknowledge that you have read the reviews. Thanks!
> >
> > -AC

---

> > > ### Comment · Reviewer_9Ssd · 2025-08-06
> > >
> > > I have read the authors' rebuttal and I think it addresses most of my questions, I'm willing to raise my score for this paper to be accepted

---

### Note · Authors · 2025-08-14

We thank all reviewers for their thoughtful comments. Overall, the reviewers note that this paper “significantly extends a set of recently published results” on learning with improvements, making a “clear contribution to the learning theory community.” The reviewers tend to agree that the paper is “well written and easy to follow” and that “helpful intuition for proofs is provided,” and we thank Reviewer 78PX for a comment regarding clarifying the statement of Theorem 3.4 to highlight the quantitative characterization via VC-dimension. Most reviewers also mentioned that the “relation to adjacent work is clearly explained, both formally and intuitively”: Reviewer 9Ssd raised some concerns with discussing related work as well as with the game-theoretic model, all of which were addressed in the rebuttal.

---

### Decision · Program_Chairs · 2025-09-17

**Decision:**

Accept (spotlight)

**Comment:**

The paper extends the new line of work in learning with improvements by providing positive learnability results on Euclidean ball improvements and also in the presence of noise.

The reviewers found the results of the paper to be interesting and significant. They also found the paper to be written clearly, which is a plus for a theory paper. Hence, I think this would be a nice contribution to NeurIPS.

For the camera-ready version, I ask the authors to fix the statement of Theorem 3.4 to add the requirement of finite VC-dimension. I would also encourage authors to add a short discussion of societal impact and incorporate some of the responses from the rebuttal to clarify the paper (particularly responses to reviewers dh1P and Zw74).